# Self-alignment of Large Video Language Models with Refined Regularized Preference Optimization

**Pritam Sarkar**
Queen's University, Canada and Vector Institute
pritam.sarkar@queensu.ca

**Ali Etemad**
Queen's University, Canada
ali.etemad@queensu.ca

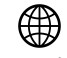 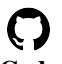 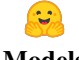

**Website**  **Code**  **Models**

## Abstract

Despite recent advances in Large Video Language Models (LVLMs), they still struggle with fine-grained temporal understanding, hallucinate, and often make simple mistakes on even simple video question-answering tasks, all of which pose significant challenges to their safe and reliable deployment in real-world applications. To address these limitations, we propose a self-alignment framework that enables LVLMs to learn from their own errors. Our proposed framework first obtains a training set of preferred and non-preferred response pairs, where non-preferred responses are generated by incorporating common error patterns that often occur due to inadequate spatio-temporal understanding, spurious correlations between co-occurring concepts, and over-reliance on linguistic cues while neglecting the vision modality, among others. To facilitate self-alignment of LVLMs with the constructed preferred and non-preferred response pairs, we introduce Refined Regularized Preference Optimization (RRPO), a novel preference optimization method that utilizes sub-sequence-level refined rewards and token-wise KL regularization to address the limitations of Direct Preference Optimization (DPO). We demonstrate that RRPO achieves more precise alignment and more stable training compared to DPO. Our experiments and analysis validate the effectiveness of our approach across diverse video tasks, including video hallucination, short- and long-video understanding, and fine-grained temporal reasoning.

## 1 Introduction

Despite recent progress in Large Video Language Models (LVLMs) [1, 2, 3, 4, 5, 6, 7, 8, 9, 10, 11, 12, 13, 14, 15], these models continue to face limitations in fine-grained temporal understanding [16, 17, 18], demonstrate a propensity for hallucination [19, 20], struggle with long-video understanding [21, 22, 23, 24, 25], and frequently make naive mistakes in short-video question-answering tasks [16, 17, 18]. Please see Figure 1 for a number of examples. These shortcomings severely limit their safe and reliable deployment in real-world applications. The underlying causes

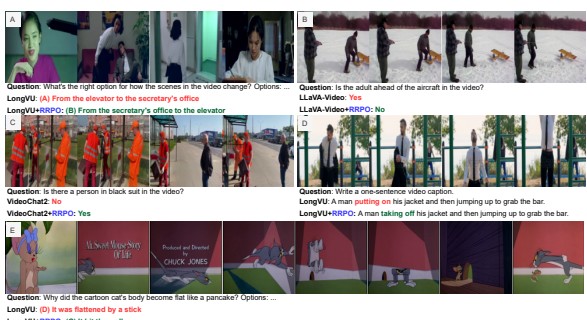

Figure 1: Examples of failure cases in simple video understanding tasks by state-of-the-art LVLMs, and improvements observed after self-alignment with RRPO.

39th Conference on Neural Information Processing Systems (NeurIPS 2025).

of these limitations are multifaceted, encompassing factors such as inadequate spatial and temporal understanding [16, 17, 18], vision-language representational misalignments [26, 22], challenges in processing long visual sequences due to context length constraints [21, 22, 23], spurious correlations between co-occurring concepts [27, 28], and an over-reliance on linguistic cues while neglecting the visual information [16, 17, 18, 29].

To address these shortcomings and enhance video understanding in LVLMs, we design a self-alignment [30] framework that enables LVLMs to learn from their own errors. Specifically, we begin by sampling video-question pairs from an open-source benchmark. We then apply spatio-temporal perturbations to the video content to mimic common errors that often arise from over-

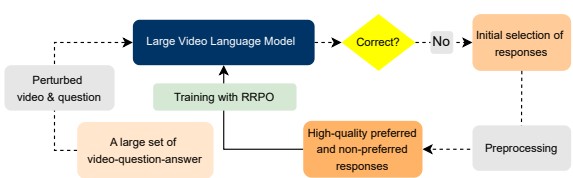

Figure 2: An overview of our self-alignment framework.

reliance on linguistic cues, spurious correlations between co-occurring concepts, and insufficient spatio-temporal understanding. Using the perturbed video and the corresponding question, we perform inference with the target LVLM. If the model's response is incorrect, we construct a self-alignment pair by treating the incorrect response as a non-preferred sample and the correct response as the preferred one, which is kept for self-alignment training. Responses that are already correct are discarded, as they offer limited self-improvement potential. Subsequently, we optimize a loss function to prioritize the preferred response over the non-preferred one. Notably, our data generation pipeline is free from human annotation and can be easily scaled. A high-level overview of our self-alignment framework is depicted in Figure 2.

To effectively train LVLMs using the generated preferred and non-preferred response pairs, we introduce *Refined Regularized Preference Optimization (RRPO)*, an approach designed to address limitations of Direct Preference Optimization (DPO) [31]. RRPO is designed to address the key drawbacks of DPO. RRPO provides a fine-grained sub-sequence-level reward to explicitly penalize the tokens containing the key concepts that are different between the preferred and non-preferred response pairs. This is contrary to DPO's response-level reward which penalizes all tokens within non-preferred responses, thus lacking precision and often proving unsuitable for fine-grained alignment. It should be noted that our fine-grained reward further benefits from computing a smaller gradient during optimization and is therefore less prone to diverge away from its initial state unlike DPO's response-level feedback along with weak regularization which can cause significant divergence from the base model, leading to suboptimal performance. However, sub-sequence-level rewards may incentivize the model to exploit shortcuts, such as outputting correct concepts without proper context or complete responses. To mitigate this, we also minimize token-wise KL-divergence [27] using a reference model on the preferred responses. This ensures that the LVLM maintains a tight bound on its likelihood across the entirety of the preferred response while reducing the likelihood on the non-preferred concepts.

Through empirical and theoretical analysis, we demonstrate that RRPO exhibits greater stability and smoother convergence during optimization compared to DPO. We validate our method on three popular LVLMs specialized for video understanding, VideoChat2, LLaVA-Video, and longVU, covering a wide range of architectures, LLMs, vision encoders, and training setups. Our in-depth evaluation demonstrates that our proposed self-alignment reduces hallucinations and improves performance in fine-grained temporal understanding, as well as in both short- and long-video understanding tasks.

In summary, our contributions are as follows:

- We design a *self-alignment* framework to facilitate self-improvement of LVLMs based on their own errors. We introduce **RRPO**, a preference optimization method that addresses the limitations of DPO by utilizing sub-sequence-level refined rewards and token-wise strong KL regularizer, resulting in more precise alignment and stable training.
- Our rigorous evaluation demonstrates the effectiveness of our proposed method across diverse video tasks, including video hallucination, short and long video understanding, and fine-grained temporal reasoning, among others. Moreover, our experimental and theoretical analyses highlight the superiority of RRPO over DPO in aligning LVLMs. We make our code, data, and model weights public to enable fast and accurate reproducibility.

## 2 Preference Responses for Self-alignment

As the first step in our framework, we construct a training dataset $\mathcal{D}$ comprising both preferred responses and responses containing incorrect concepts in order to align the LVLM $\pi_\theta$ to favor correct concepts over incorrect ones. We begin by utilizing a large and diverse publicly available video instruction tuning dataset, containing triplets of video $x_v$, question $x_q$, and their human-preferred answers $y^+$. To generate non-preferred responses $y^-$, we obtain perturbed videos $\hat{x}_v = f_p(x_v)$, where $f_p$ is a perturbation function which masks a large portion of frames and applies temporal shuffling, compromising spatio-temporal consistency. Our intention from this step is to provoke the LVLM to generate responses primarily based on language cues or their parametric knowledge. These perturbed videos $\hat{x}_v$ are fed to $\pi_\theta$ as inputs to generate responses with potential erroneous concepts $y^- = \pi_\theta(x_q, \hat{x}_v)$. An example is illustrated in Figure 3. We then verify the correctness of $y^-$, retaining incorrect responses and discarding correct ones. Next, We employ an LLM to meticulously com-

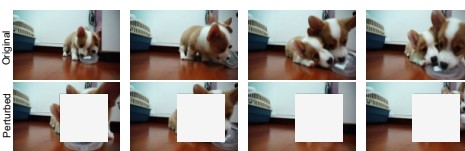

Figure 3: An example of perturbed video.

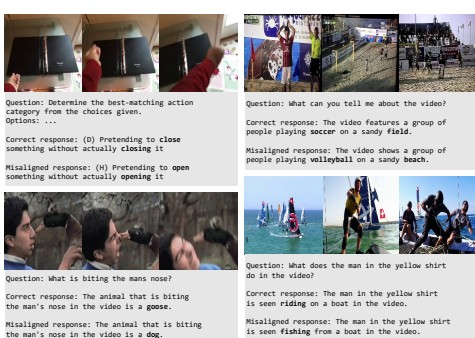

Figure 4: A few training samples. We bold the correct and incorrect concepts.

pare $y^-$ against $y^+$ and identify key incorrect concepts within $y^-$, ensuring that for each incorrect concept $y_i^- \in y^-$ there exists a corresponding correct concept $y_i^+ \in y^+$. In the context of our work, a concept can be actions, objects, attributes, relations, and other key elements in the response that contribute to the semantic understanding of the video. Furthermore, for lengthy responses, we maintain structural similarity between the incorrect and correct versions by rewriting the correct response while incorporating the incorrect concepts. Finally, we apply deduplication based on these correct-incorrect concept pairs, constructing a high-quality training dataset. Examples are provided in Figure 4.

## 3 RRPO

**Preliminaries.** Given an input $x = \{x_q, x_v\}$ with a pair of responses $\{y^+, y^-\}$, where $y^+ \succ y^-|x$, DPO [31] can be employed to align $\pi_\theta$ to favor $y^+$ over $y^-$. This is achieved by maximizing the reward margin between $\pi_\theta$ and a reference model $\pi_{\text{ref}}$, using the following training objective:

$$\mathcal{L}_{\text{DPO}}(\pi_\theta; \pi_{\text{ref}}) = -\mathbb{E}_{(x,y^+,y^-)\sim\mathcal{D}}\left[\log\sigma\left(r_\theta(x, y^+) - r_\theta(x, y^-)\right)\right], \quad (1)$$

where reward $r_\theta(x, y) = \beta\log\frac{\pi_\theta(y|x)}{\pi_{\text{ref}}(y|x)}$ and $\beta$ controls the deviation from $\pi_{\text{ref}}$. Note that $r_\theta(x, y)$ is calculated at a response-level by penalizing all the tokens in $y$, despite the fact that the difference between $y^+$ and $y^-$ might only be limited to a *few* key tokens. Such response-level reward can be considered coarse-grained reward modeling and is not suitable for fine-grained alignment. Moreover, as the reward is calculated by penalizing all the tokens in the response, the gradient for $\mathcal{L}_{\text{DPO}}$ tends to be very large for long responses, and accordingly $\pi_\theta$ can diverge to an undesired state thus losing its out-of-the-box capabilities [27, 32, 33].

**Refined Regularized Preference Optimization.** Our goal is to design a method that can provide a fine-grained reward to penalize individual sub-sequences consisting of the tokens belong to the key differing concepts between $y^+$ and $y^-$. We refer to this as refined reward modeling. Let $y$ be expressed as a sequence of tokens $T = \{t_1, t_2, \ldots, t_{|y|}\}$. Here, the $i$-th sub-sequence $y_i$ can be expressed as $T[s_i : e_i]$, where $s_i$ and $e_i$ are the start and end indices of $y_i$ with $1 \leq s_i \leq e_i \leq |y|$. Therefore, the reward for $y_i$ can be computed as:

$$r_\theta(x, y_i) = \beta\log\frac{\prod_{j=s_i}^{e_i}\pi_\theta(t_j|x, t_{<j})}{\prod_{j=s_i}^{e_i}\pi_{\text{ref}}(t_j|x, t_{<j})}. \quad (2)$$

Assuming there exists $N$ such sub-sequences, we can train $\pi_\theta$ to maximize the total reward margin

$$u = \sum_{i=1}^{N} u_i = \sum_{i=1}^{N} \big( r_\theta(x, y_i^+) - r_\theta(x, y_i^-) \big). \tag{3}$$

Subsequently, we replace the reward formulation in Equation 1 with our refined reward modeling as:

$$\mathcal{L}_{\text{RRPO}}^{(\text{rank})}(\pi_\theta; \pi_{\text{ref}}) = -\mathbb{E}_{(x, y^+, y^-) \sim \mathcal{D}} \Big[ \log \sigma(u) \Big]. \tag{4}$$

However, the sparsity of the rewards used to calculate $\mathcal{L}_{\text{RRPO}}^{(\text{rank})}$ may allow $\pi_\theta$ to exploit shortcuts and effectively game the reward function by merely outputting key concepts without appropriate context or generating complete responses. This *reward hacking* may occur since $\pi_\theta$ is incentivized to maximize reward margin based on the differing sub-sequences; the model can learn to produce short, incomplete responses that contain the correct key concepts, even if those responses lack overall coherence, fluency, or completeness. To mitigate this reward hacking, we introduce a regularizer between $\pi_\theta$ and $\pi_{\text{ref}}$, based on token-wise KL (TKL) divergence [27, 34], as follows:

$$\mathbb{D}_{\text{TKL}}\big(x, y; \pi_{\text{ref}} \| \pi_\theta\big) = \sum_{t=1}^{|y|} \mathbb{D}_{\text{KL}}\big(\pi_{\text{ref}}(\cdot \mid [x, y_{<t}]) \| \pi_\theta(\cdot \mid [x, y_{<t}])\big). \tag{5}$$

Accordingly, to ensure $\pi_\theta$ retains its original likelihood across the entirety of $y^+$ while reducing the likelihood of non-preferred concepts, we optimize $\mathbb{D}_{\text{TKL}}$ between $\pi_\theta$ and $\pi_{\text{ref}}$ over $y^+$. We then modify Equation 4 and define the final RRPO loss as:

$$\mathcal{L}_{\text{RRPO}}(\pi_\theta; \pi_{\text{ref}}) = -\mathbb{E}_{(x, y^+, y^-) \sim \mathcal{D}} \Big[ \log \sigma(u) + \alpha \cdot \mathbb{D}_{\text{TKL}}\big(x, y^+\big) \Big], \tag{6}$$

where $\alpha$ controls the divergence between $\pi_\theta$ and $\pi_{\text{ref}}$, and $\pi_{\text{ref}}$ is kept fixed.

**How is RRPO update different from DPO?** The gradient for $\mathcal{L}_{\text{RRPO}}$ can be obtained as:

$$\nabla_\theta \mathcal{L}_{\text{RRPO}} = -\nabla_\theta \mathbb{E} \Big[ \log \sigma(u) + \alpha \cdot \mathbb{D}_{\text{TKL}}\big(x, y^+\big) \Big] = -\mathbb{E} \Big[ \nabla_\theta \log \sigma(u) + \alpha \cdot \nabla_\theta \mathbb{D}_{\text{TKL}}\big(x, y^+\big) \Big]. \tag{7}$$

First, let's calculate the gradient for the ranking loss. Using chain rule, we can write $\nabla_\theta \log \sigma(u) = \Big[ \frac{\sigma'(u)}{\sigma(u)} \nabla_\theta u \Big]$. Using the identity $\sigma'(u) = \sigma(u)(1 - \sigma(u))$, we have $\frac{\sigma'(u)}{\sigma(u)} = 1 - \sigma(u) = \sigma(-u)$. Therefore, $\nabla_\theta \log \sigma(u) = \Big[ \sigma(-u) \nabla_\theta u \Big]$. Recalling our sub-sequence-level reward modeling defined in Equation 3, we can obtain:

$$\nabla_\theta u = \sum_{i=1}^{N} \nabla_\theta u_i = \beta \sum_{i=1}^{N} \Big( \sum_{j=s_i^+}^{e_i^+} \nabla_\theta \log \pi_\theta(t_j^+ | x, t_{<j}^+) - \sum_{j=s_i^-}^{e_i^-} \nabla_\theta \log \pi_\theta(t_j^- | x, t_{<j}^-) \Big), \tag{8}$$

as $\pi_{\text{ref}}$ is fixed. Assuming the norm of the gradient of the log-probability for any single token is bounded, $\|\nabla_\theta \log \pi_\theta(t_j | x, t_{<j})\| \leq M$ for some $M > 0$, and that each differentiating sub-sequence $y_i^+$ or $y_i^-$ has an average length $L$, we can further bound the gradient norm of the ranking loss:

$$\|\nabla_\theta \mathcal{L}_{\text{RRPO}}^{(\text{rank})}\| \leq \mathbb{E} \Big[ \sigma(-u) \|\nabla_\theta u\| \Big] \leq \beta \sum_{i=1}^{N} (M L_i^+ + M L_i^-) \approx \beta M (2NL), \tag{9}$$

where $L$ is the average length of the sub-sequences ($L \approx L_i^+ \approx L_i^-$). In contrast, the DPO gradient involves sums over the *entire* lengths of $y^+$ and $y^-$. A similar analysis for DPO yields a bound proportional to the total lengths:

$$\|\nabla_\theta \mathcal{L}_{\text{DPO}}\| \leq \beta M (|y^+| + |y^-|). \tag{10}$$

Crucially, the total length of the differentiating sub-sequences is typically much smaller than the total length of the full responses, i.e., $2NL \ll |y^+| + |y^-|$. Therefore, the upper bound on the gradient magnitude stemming from the ranking objective is smaller for RRPO compared to DPO:

$$\|\nabla_\theta \mathcal{L}_{\text{RRPO}}^{(\text{rank})}\| \ll \|\nabla_\theta \mathcal{L}_{\text{DPO}}\|.$$

Now, let's consider the term $\nabla_\theta \mathbb{D}_{\text{TKL}}(\cdot)$ in Equation 7. Based on the formulation of $\mathbb{D}_{\text{KL}}(\pi_{\text{ref}} || \pi_\theta) = \sum_a \pi_{\text{ref}}(a) \log \frac{\pi_{\text{ref}}(a)}{\pi_\theta(a)}$ where $a$ represents a token in the vocabulary, and since $\pi_{\text{ref}}$ fixed, we derive:

$$\nabla_\theta \mathbb{D}_{\text{TKL}}(x, y^+) = \sum_{t=1}^{|y^+|} \nabla_\theta \mathbb{D}_{\text{KL}}(\cdot) = -\sum_{t=1}^{|y^+|} \sum_a \pi_{\text{ref}}(a \mid x, y^+_{<t}) \nabla_\theta \log \pi_\theta(a \mid x, y^+_{<t}). \quad (11)$$

Note that $\nabla_\theta \mathbb{D}_{\text{TKL}}$ is always negative. Therefore, for $\alpha > 0$ in Equation 7, the gradient magnitude of $\mathcal{L}_{\text{RRPO}}$ is further reduced than $\mathcal{L}_{\text{RRPO}}^{(\text{rank})}$.

$$\|\nabla_\theta \mathcal{L}_{\text{RRPO}}\| < \|\nabla_\theta \mathcal{L}_{\text{RRPO}}^{(\text{rank})}\| < \|\nabla_\theta \mathcal{L}_{\text{DPO}}\|$$

This mathematical derivation confirms that the proposed RRPO loss function effectively reduces the gradient, as initially hypothesized. The reduced gradient of RRPO ensures more stable updates compared to DPO in gradient-based optimization while simultaneously enabling precise penalties on specific tokens without the risk of significant divergence. Furthermore, the $\mathbb{D}_{\text{TKL}}$ term acts as a trust-region constraint [35], preventing the model from making large, destabilizing updates. As a result, RRPO allows larger learning rates and yielding smoother convergence in practice. We present the pseudocode of RRPO in Appendix A.

## 4 Experiment Setup

**Base models.** We use VideoChat2$_{7B}$ [3], LLaVA-Video$_{7B}$ (also known as LLaVA-Next-Video$_{7B}$) [9], and LongVU$_{7B}$ [1] as our base models. These models are carefully selected to evaluate our method across diverse LLM architectures, vision encoders, cross-modal adapters, and training setups. For instance, among these models, VideoChat2 employs a video encoder, while the others rely on image encoders. Additionally, LongVU incorporates two vision encoders, whereas the rest use a single vision encoder. VideoChat2 utilizes QFormer [36] as its cross-modal adapter, whereas LLaVA-Video employs MLP projection layers. These models further differ in their LLM architectures, context lengths, and training setups, among other aspects.

**Training data.** Based on the availability and diversity of video-language instructions, we use VideoChat-IT [3] as our primary source for training samples. Specifically, we select a subset of VideoChat-IT encompassing eight video datasets: Kinetics700 [37], Something-Something-v2 [38], VideoChat [39], VideoChatGPT [40], CLEVRER [41], NEXTQA [42], EgoQA [43], and TGIF [44]. These datasets span a range of tasks, including video description, question answering, reasoning, and conversation. For the perturbation step, we mask a significant portion (25%-50%) of each frame and shuffle the temporal order. We explore three types of temporal perturbations: (*i*) random shuffling, (*ii*) local shuffling, and (*iii*) global shuffling. In random shuffling, frames are shuffled arbitrarily across time. For local shuffling, frames are initially segmented into smaller chunks, and the frames within each chunk are then shuffled. In global shuffling, the order of these chunks is shuffled, rather than individual frames. During inference, based on the LVLMs' input capacity, we utilize a maximum of 16, 64, and 100 frames for VideoChat2, LLaVA-Video, and LongVU, respectively.

Following the generation of the responses, we verify their correctness. For Multiple Choice Questions (MCQ) and Binary Question Answering (BinaryQA) tasks, verification is straightforward, using regex-based checks. However, for open-ended questions, this method proves inadequate, as semantically equivalent responses can be expressed in diverse phrasings. Consequently, for open-ended questions, we employ a powerful LLM, GPT-4o-mini [45], as a judge [46, 47, 48, 49], to ascertain correctness by comparing the

Table 1: Key statistics of the generated training samples.

| Model | # Samples | Unique pairs |
|---|---|---|
| VideoChat2$_{7B}$ | 25K | 18K |
| LongVU$_{7B}$ | 22K | 14K |
| LLaVA-Video$_{7B}$ | 21K | 16K |

generated response with the ground truth from the video instruction tuning dataset. Additionally, for long responses, we employ GPT-4o-mini to rewrite the correct response while incorporating the incorrect concepts from the generated response. This ensures that correct and incorrect concepts are aligned across both preferred and non-preferred responses. The key statistics of our training data are presented in Table 1. The prompts used during pre-processing with GPT-4o-mini are in Appendix D.

**Implementation details.** For all base models, we utilize LoRA [50] for training, applying it specifically to the LLMs while freezing all other parameters. Unless otherwise specified, we utilize

16 frames for self-alignment training; the rest follows the default training setup of each respective base model. We use $4\times$ A100 80GB GPUs for training, with the training time varying between 1 to 10 hours. Additional implementation details and hyperparameters are provided in Appendix E.

**Evaluation benchmarks.** To assess the impact of our self-alignment framework, we conduct evaluations across a diverse range of video understanding tasks. Specifically, we choose TVBench [17] and TempCompass [20] for fine-grained temporal understanding, VideoHallucer [19] and VidHalluc [51] for video hallucination, MVBench [3] and VideoMME [52] for short video understanding, and MLVU [24] and LongVideoBench [25] for long video understanding. It should be noted that these benchmarks, while selected for specific tasks, often assess overlapping video understanding capabilities. For instance, while VideoHallucer and VidHalluc are primarily used for hallucination detection, they also evaluate temporal grounding [19, 51]. Similarly, while TVBench mainly focuses on temporal understanding, it covers short video understanding as well [17]. Given its inclusion of short, medium, and long videos, VideoMME explores video understanding of varying lengths.

## 5 Results and Analysis

This section details our experimental evaluation, which encompasses a comprehensive and comparative analysis against other alignment methods and existing off-the-shelf aligned LVLMs. Furthermore, we present a detailed analysis focusing on the trade-offs between post-alignment divergence and performance. To gain deeper insights into the impact of our proposed method, we conduct experiments exploring its effects on fine-grained temporal understanding, hallucination mitigation, and performance on comprehensive video understanding of varying lengths. Finally, our evaluation includes an investigation into the influence of data, the scaling of input frames, and the presence of subtitles on the performance of our method. We conclude with a discussion regarding the generalization capabilities of our approach.

Table 2: Comparison with existing preference optimization methods and RRPO ablation variants. $\Delta = \frac{1}{N} \sum (\text{acc}_{\text{aligned}} - \text{acc}_{\text{base}})$ and $\%\Delta = \frac{100}{N} \sum (\text{acc}_{\text{aligned}} - \text{acc}_{\text{base}}) / \text{acc}_{\text{base}}$, where $N$ is the number of evaluation benchmarks used for each ablation. RRPO consistently outperforms existing alignment methods.

| | TVB | VHall | VMME | MLVU | $\Delta/\%\Delta$ |
|---|---|---|---|---|---|
| **LongVU$_{7B}$ (base)** | 53.7 | 39.2 | 56.2 | 63.6 | — |
| + DPO [31] | 54.3 | 40.9 | 56.6 | 63.6 | 0.7/1.5 |
| + DPA [27] | 54.6 | 40.3 | 56.9 | 63.9 | 0.7/1.5 |
| + TDPO [34] | 53.9 | 41.4 | 57.0 | 63.8 | 0.8/1.9 |
| + DDPO [53] | 54.2 | 41.7 | 56.7 | 63.6 | 0.9/2.0 |
| + RRPO w/o R$^*$ | 54.3 | 43.0 | **57.8** | 64.5 | 1.7/3.8 |
| + RRPO w/o $\mathbb{D}_{\text{TKL}}$ | 54.9 | 39.1 | 57.4 | 63.9 | 0.6/1.1 |
| + **RRPO (ours)** | **56.5** | **44.0** | 57.7 | **64.5** | **2.5/5.4** |

*( Abbreviations, TVB: TVBench; VHall: VideoHallucer; VMME: VideoMME)*

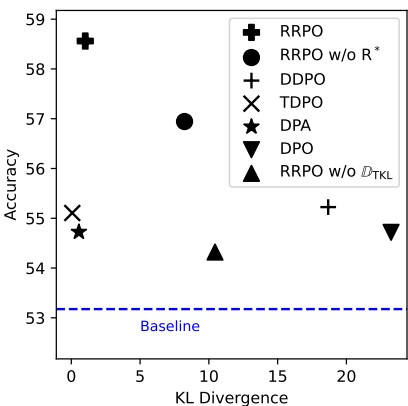

Figure 5: Performance relative to model divergence. RRPO exhibits the best performance-divergence trade-off.

**Q1. How well does RRPO perform compared to other alignment methods?** We conduct an in-depth analysis of RRPO against other recent preference optimization methods designed to address DPO's limitations, namely TDPO [34], DDPO [53], and DPA [27] and present the results in Table 2. More specifically, DDPO was introduced to provide fine-grained rewards, TDPO to enhance DPO's regularization, and DPA to address both of these challenges. A detailed discussion of their objectives, highlighting similarities and differences, is provided in Appendix B. Furthermore, we include two ablation variants of RRPO: one without the refined reward (RRPO w/o R$^*$) and the other without the token-wise KL regularizer (RRPO w/o $\mathbb{D}_{\text{TKL}}$). Our study covers various aspects of video understanding, including fine-grained temporal understanding on TVBench, hallucination mitigation on VideoHallucer, comprehensive video understanding on VideoMME, and long video understanding on MLVU. The results presented in Table 2 demonstrate that RRPO consistently outperforms DPO, DPA, TDPO, and DDPO across all benchmarks. Moreover, among the ablation variants of RRPO, the inclusion of $\mathbb{D}_{\text{TKL}}$ alone yields significant performance gains, which are further enhanced by incorporating the refined reward. We present qualitative comparisons in Appendix G.

Table 3: Comparison with off-the-shelf aligned LVLMs. Ours LLaVA-Video-RRPO outperforms LLaVA-Video-TPO across all setups, both of which are based on LLaVA-Video$_{7B}$.

| Model | TV Bench | TempCo-mpass$_{Avg}$ | Video Hallucer | Vid Halluc | MV Bench | Video MME | MLVU$_{val}$ (M-Avg) | LongVideo Bench$_{Val}$ |
|---|---|---|---|---|---|---|---|---|
| LLaVA-Video-TPO [54] | 51.1 | 66.1 | 50.6 | 76.3 | 60.6 | **65.6**/71.5 | 68.7 | 60.1 |
| LLaVA-Video-RRPO (ours) | **52.2** | **67.4** | **55.8** | **76.6** | **62.0** | 65.5/**71.8** | **69.4** | **61.3** |

Table 4: Evaluating our self-aligned LVLMs on diverse video understanding benchmarks. We **bold** the best in each group. 16f and 32f indicate the number of frames utilized during training. #F indicates number of frames used during inference. Here, we presents the overall results averaged across sub-categories where applicable, with detailed results available in Appendix C.

| Models | #F | TV Bench | TempCo-mpass$_{Avg}$ | Video Hallucer | Vid Halluc | MV Bench | Video MME | MLVU$_{val}$ (M-Avg) | LongVideo Bench$_{Val}$ |
|---|---|---|---|---|---|---|---|---|---|
| Video-LLaVA$_{7B}$ [10] | | 33.8 | 49.9 | 17.8 | 40.3 | 42.5 | 39.9 | 29.3 | 39.1 |
| VideoLLaMA2$_{7B}$ [5] | | 42.9 | 43.4 | 10.0 | 66.3 | 54.6 | 62.4 | 48.4 | – |
| LongVA$_{7B}$ [21] | | – | 57.0 | – | – | – | 52.6 | 42.1 | – |
| MiniCPM-V 2.6$_{7B}$ [55] | | – | 66.3 | – | – | – | 60.9 | – | 54.9 |
| NVILA$_{7B}$ [56] | | – | – | – | – | 68.1 | 64.2 | 70.1 | 57.7 |
| LongVILA$_{7B}$ [8] | | – | – | – | – | 67.1 | 60.1 | – | 57.1 |
| Qwen2-VL$_{7B}$ [57] | | 43.8 | 67.9 | – | – | 67.0 | 63.3 | 65.5 | 55.6 |
| LLaVA-NeXT-Video-DPO$_{7B}$ [58] | | – | 53.8 | 32.0 | – | – | – | – | 43.5 |
| ShareGPT4Video$_{8B}$ [59] | | – | 61.5 | 15.8 | 30.9 | – | 39.9 | 34.2 | 39.7 |
| PLLaVA$_{13B}$ [60] | | – | – | 41.2 | 48.2 | – | – | – | 45.6 |
| Aria$_{8x3.5B}$ [61] | | 51.0 | 69.6 | – | – | 69.7 | 67.6 | – | 63.0 |
| LLaVA-NeXT-Video-DPO$_{34B}$ [58] | | – | – | 32.3 | 49.5 | – | – | – | 50.5 |
| Qwen2-VL$_{72B}$ [57] | | 52.7 | – | – | – | 73.6 | 71.2 | – | – |
| GPT-4o [62] | | 39.9 | 73.8 | 53.3 | 81.2 | 49.1 | 71.9 | 54.5 | 66.7 |
| Gemini 1.5 Pro [11] | | 47.6 | 67.1 | 37.8 | 72.8 | 60.5 | 75.0 | – | 64.0 |
| *(The above models are presented for reference only, and may not be suitable for direct comparisons.)* | | | | | | | | | |
| VideoChat2$_{7B}$ [3] | | – | 50.8 | 7.8 | – | 60.4 | 39.5 | – | 39.3 |
| VideoChat2$_{7B}$ | 16 | 44.0 | 59.3 | 23.1 | 73.3 | **60.2** | 41.0 | 46.4 | 40.4 |
| + DPO (16f) | 16 | 45.7 | 60.0 | 22.1 | 72.4 | 59.6 | 43.0 | 47.4 | 41.0 |
| + RRPO (16f) (Ours) | 16 | **45.8** | **60.2** | **32.9** | **76.4** | 59.0 | **44.3** | **47.9** | **42.8** |
| LLaVA-Video$_{7B}$ [9] | | 45.6 | – | – | – | 58.6 | 63.3 | 70.8 | 58.2 |
| LLaVA-Video$_{7B}$ | 64 | 51.0 | 66.0 | 50.0 | **76.6** | 61.1 | 64.0 | 68.6 | 60.1 |
| + DPO (16f) | 64 | 51.9 | 66.4 | 53.3 | 76.5 | 60.6 | 63.1 | 67.4 | 59.4 |
| + RRPO (16f) (Ours) | 64 | 51.9 | 66.8 | 55.7 | 76.5 | 62.2 | 64.5 | 69.1 | **60.4** |
| + RRPO (32f) (Ours) | 64 | **52.2** | **67.4** | **55.8** | **76.6** | 62.1 | 64.5 | **69.4** | 60.1 |
| LongVU$_{7B}$ [1] | | – | – | – | – | 66.9 | – | 65.4 | – |
| LongVU$_{7B}$ | 1fps | 53.7 | 63.9 | 39.2 | 67.3 | 65.5 | 56.2 | 63.6 | 48.6 |
| + DPO (16f) | 1fps | 54.3 | 64.3 | 40.9 | 68.5 | 65.9 | 56.6 | 63.6 | 49.4 |
| + RRPO (16f) (Ours) | 1fps | **56.5** | **64.5** | **44.0** | **71.7** | **66.8** | **57.7** | **64.5** | **49.7** |

*For transparency, we report base LVLMs' results from the literature where available, marked in grey. Differences from our reproduced results, both higher and lower, stem from variations in frame count, input prompt, GPT variant used for evaluation, and occasional implementation issues. For fairness, self-aligned variants are compared against our reproduced results.*

**Q2. How much does the model diverge post alignment?** Understanding the extent of model divergence after alignment is essential to ensure that the process refines behavior without undermining core capabilities as excessive divergence can lead to instability, reduced generalization, and loss of valuable pre-trained knowledge. Figure 5 presents a comparative analysis of performance improvements against model divergence following preference optimization. Despite using a $10\times$ higher learning rate (which is facilitated through the smaller and more stable gradient updates), RRPO exhibits a KL divergence of 1 compared to DPO's KL divergence of 20. While TDPO and DPA exhibit almost the same divergence as RRPO, their performance across all evaluation benchmarks is substantially worse. RRPO, in contrast, demonstrates the optimal performance-divergence trade-off.

**Q3. How does RRPO performance compared to off-the-shelf aligned LVLMs?** We further evaluate our aligned LVLM against other off-the-shelf aligned LVLMs. Specifically, we compare our RRPO-trained LLaVA-Video with the concurrent work LLaVA-Video-TPO, both of which are based on the LLaVA-Video-Qwen$_{7B}$ [9]. LLaVA-Video-TPO is trained using a combination of DPO and SFT with a manually curated video-language dataset. The results in Table 3 demonstrate that RRPO is significantly more effective than concurrent methods. Our LLaVA-Video-RRPO outperforms LLaVA-Video-TPO across all setups, with performance gains of up to $5.2\%$.

Table 5: Impact of different spatio-temporal perturbations in generating non-preferred responses. Default perturbations for each model are colored.

| | TVB | VHall | VMME | MLVU | Δ/%Δ |
|---|---|---|---|---|---|
| **VideoChat2$_{7B}$ (base)** | 44.0 | 23.1 | 41.0 | 46.4 | – |
| + None | 40.7 | 16.0 | 39.9 | 43.8 | -3.5/-11.6 |
| + RS | 43.0 | 23.3 | 41.8 | 46.3 | 0.0/0.1 |
| + Mask | 44.0 | 26.2 | 43.4 | 48.8 | 2.0/6.1 |
| + LS-Mask | 44.6 | 28.2 | 44.6 | 49.4 | 3.1/9.7 |
| + GS-Mask | 44.2 | 28.8 | 44.1 | 46.3 | 2.2/8.1 |
| + RS-Mask | 45.8 | 32.9 | 44.3 | 47.9 | **4.1/14.4** |
| **LLaVA-Video$_{7B}$ (base)** | 51.0 | 50.0 | 64.0 | 68.6 | – |
| + LS-Mask | 51.9 | 55.7 | 64.5 | 69.1 | **1.9/3.7** |
| + GS-Mask | 51.3 | 54.8 | 64.2 | 68.7 | 1.4/2.7 |
| + RS-Mask | 51.5 | 51.6 | 64.6 | 69.6 | 0.9/1.6 |
| **LongVU$_{7B}$ (base)** | 53.7 | 39.2 | 56.2 | 63.6 | – |
| + LS-Mask | 56.5 | 44.0 | 57.7 | 64.5 | **2.5/5.4** |
| + GS-Mask | 55.0 | 43.9 | 56.9 | 64.5 | 1.9/4.3 |
| + RS-Mask | 55.1 | 42.4 | 57.0 | 64.3 | 1.5/3.3 |

Table 6: Impact of data size.

| | TVB | VHall | VMME | MLVU | Δ/%Δ |
|---|---|---|---|---|---|
| Baseline | 51.0 | 50.0 | 64.0 | 68.6 | – |
| + 5K | 50.9 | 53.7 | 64.0 | 69.0 | 1.0/1.9 |
| + 10K | 51.2 | 53.8 | 64.3 | 69.0 | 1.2/2.3 |
| + 15K | 51.8 | 54.4 | 64.2 | 68.9 | 1.4/2.8 |
| + 20K | 51.9 | 55.7 | 64.5 | 69.1 | **1.9/3.7** |

Table 7: Impact of varying the number of frames at inference.

| | | TVB | MVB | LVB |
|---|---|---|---|---|
| 32 | Baseline | 49.6 | 61.2 | 58.4 |
| | + RRPO | 51.3 | 61.7 | 58.9 |
| 64 | Baseline | 51.0 | 61.1 | 60.1 |
| | + RRPO | **52.2** | **62.1** | 60.1 |
| 128 | Baseline | 49.4 | 60.5 | 60.3 |
| | + RRPO | 51.3 | 61.2 | **61.3** |

Table 8: Impact of using subtitles along with videos (VMME).

| | without | with |
|---|---|---|
| VideoChat2$_{7B}$ | 41.0 | 48.0 |
| + RRPO | **44.3** | **49.4** |
| LLaVA-Video$_{7B}$ | 63.8 | 67.4 |
| + RRPO | **64.5** | **68.0** |
| LongVU$_{7B}$ | 56.2 | 62.0 |
| + RRPO | **57.7** | **63.1** |

*( Abbreviations, RS: Random Shuffle; LS: Local Shuffle; GS: Global Shuffle; TVB: TVBench; VHall: VideoHallucer; VMME: VideoMME; LVB: LongVideoBench.)*

**Q4. Does our method improve fine-grained temporal understanding?** To evaluate this, we utilize TVBench [17] and TempCompass [20], designed to test the temporal understanding abilities of LVLMs. TVBench tests capabilities across various temporal tasks, including action localization, directional movement, and scene transitions, among others. Similarly, TempCompass evaluates performance on video captioning, caption matching, MCQ, and BinaryQA, covering video understanding tasks such as event ordering, action identification, and state change. As shown in Table 4, our method, RRPO, consistently improves base model performance by up to 2.8%, demonstrating its effectiveness in enhancing fine-grained temporal understanding and outperforming DPO across all setups.

**Q5. Does our method effectively mitigate hallucinations?** Hallucination occurs when LVLMs produce responses that are ungrounded, referred to as intrinsic hallucination, or unverifiable, referred to as extrinsic hallucination. Hallucination presents a significant obstacle to the reliable use of LVLMs. To evaluate the impact of our method on video hallucination, we employ VideoHallucer [19] and VidHalluc [51]. VideoHallucer tests LVLMs for both extrinsic and intrinsic hallucinations while including both spatial and temporal hallucinations. Additionally, VidHalluc focuses specifically on temporal hallucinations, such as action hallucination. As shown in Table 4, RRPO significantly reduces hallucination across all base models. Specifically, RRPO improves performance by 4.8% to 8.8% on VideoHallucer and by up to 4.4% on VidHalluc. In most cases, RRPO demonstrates a substantial performance advantage over DPO, with gains reaching 10.8%.

**Q6. Does our method improve comprehensive video understanding across varying video lengths?** To evaluate the comprehensive video understanding capabilities of LVLMs across varying video lengths, we leverage four benchmarks: MVBench [3], VideoMME [52], MLVU [24], and LongVideoBench [25]. Together, these benchmarks span a wide variety of perception and reasoning tasks, focusing on objects, actions, their attributes, and holistic video understanding. Among these, MVBench is specifically designed for a comprehensive evaluation of *short videos*, while MLVU and LongVideoBench offer thorough evaluations for *long videos*. VideoMME provides a comprehensive assessment across videos of *varying lengths*. As shown in Table 4, consistent improvements are observed across all benchmarks for LongVU and LLaVA-Next. For VideoChat2, self-alignment leads to substantial gains in three out of four setups, with only a minor regression in MVBench. Furthermore, RRPO consistently outperforms DPO, further demonstrating the advantages of fine-grained alignment in enhancing comprehensive video understanding.

**Q7. How do perturbations in our data generation pipeline impact the quality of non-preferred responses?** To assess the impact of the perturbations on the quality of non-preferred responses, we conduct in-depth analyses and present the results in Table 5. Our key observations are as follows: First, non-preferred responses generated without video perturbations leads to diminished model performance, likely due to reduced generalizability. Second, temporal perturbations alone are ineffective, although their combination with masking significantly boosts performance. Among the spatio-temporal perturbations, random shuffling with masking (RS-Mask) improves performance for models processing fewer frames (e.g., VideoChat2) whereas local shuffling with masking (LS-Mask) proves superior for models handling longer sequences (e.g., LongVU, LLaVA-Video).

**Q8. How does our method scale with training data?** To test the impact of scaling the amount of data, we perform an experiment by varying the number of training samples from 5K to 20K, incrementing by 5K. As shown in Table 6, performance improves with data size. This suggests that our data-generation pipeline is effective in producing high-quality training samples for self-alignment.

**Q9. How does performance vary with the number of input frames?** We investigate the effect of scaling the number of visual input frames during both inference and self-alignment training. For inference, we evaluate LLaVA-Video using 32, 64, and 128 frames across TVBench, MVBench, and LongVideoBench. The results are presented in Table 7. Our key observations are as follows: First, RRPO consistently improves performance over the base model across all setups. Second, neither the base model nor RRPO exhibits performance gains beyond 64 frames on TVBench and MVBench. This is likely due to the short-video nature of these benchmarks, where higher frame counts result in redundant frame resampling. However, for long-video understanding, increasing frame counts yields performance improvements, particularly for RRPO with a $1.2\%$ improvement compared to the base model's $0.2\%$ gain. Subsequently, we explore the impact of increasing the number of frames during self-alignment training. Specifically, we raise the frame count from our default 16 to 32. The results presented in Table 4 demonstrate that this increase enhances performance. Notably, we observe a consistent performance improvement on fine-grained temporal understanding tasks, as evidenced by the gains on TVBench and TempCompass.

**Q10. Does our method retain its performance advantage with subtitles?** Subtitles generally enhance video understanding by providing additional language cues that LVLMs can leverage. Thus, we investigate whether our method maintains its benefits over base models when subtitles are included. As shown in Table 8, our method demonstrates consistent improvements across all setups.

**Q11. Does our method generalize across diverse LVLM architectures and training setups?** Given the rapid evolution of LVLMs, we carefully select VideoChat2, LLaVA-Video, and LongVU as the base models to cover a wide variety of design choices and training methodologies (e.g., Table 4). Specifically, VideoChat2 uses the UMT [63] video encoder, while LLaVA-Video and LongVU use DINOv2 [64] and SigLIP [65] as their image encoders. On the other hand, LongVU employs dual vision encoders, unlike the others. Moreover, the cross-modal adapters range from query-based for VideoChat2 to MLP projections for LLaVA-Video, and a combinations of both in the case of LongVU. Furthermore, the overlap between self-alignment training samples and instruction tuning data differs across models, with VideoChat2 having the highest overlap and LLaVA-Video having the least. Importantly, our self-alignment consistently improves performance across these diverse setups, even when reusing instruction tuning data.

## 6 Related work

Recent years have seen a surge in the development of LVLMs with improved video understanding capabilities [1, 2, 3, 4, 5, 6, 7, 8, 9, 10, 11, 12, 13, 14, 15]. This progress can be attributed to several key factors: (1) the development of diverse video benchmarks [3, 7, 66, 43, 67], which enable LVLMs to follow human instructions and reason across a variety of video tasks; (2) architectural innovations that support the use of rich, dense visual features and enable efficient processing of long sequences [21, 8, 1, 68, 69, 70, 5, 71]; and (3) advancements in training algorithms for both the pre-training [72, 63, 65, 73] and post-training [74, 75, 31] stages. LVLM training generally involves multi-stage processes [76, 77], with pre-training typically focusing on representational alignment between video and language [3, 78, 36], and post-training refining the model's ability to follow instructions [3, 78, 79], reduce hallucination [53, 27], improve reasoning skills [80, 81], and align the model with human preferences [80, 81]. Our introduced RRPO is a post-training alignment method designed to enhance the overall video understanding capabilities of LVLMs. While concurrent works [82, 83, 54] have explored post-training alignments of LVLMs, they directly adopt DPO [31], which proves ineffective in fine-grained alignment, as discussed in Section 5.

## 7 Conclusion

To improve the video understanding abilities of LVLMs, we design a self-alignment framework that enables LVLMs to learn from their own errors. These errors commonly occur due to their lack of spatio-temporal reasoning, over-reliance on linguistic cues, and spurious correlations between

co-occurring concepts, among others. To effectively align LVLMs against such errors, we introduce RRPO, a novel preference optimization method designed for fine-grained alignment through refined reward modeling with strong regularization. Our in-depth experiments and analyses show that RRPO training is more stable and highly effective compared to prior and concurrent preference optimization methods across diverse tasks. Moreover, the fine-grained reward modeling in RRPO improves capabilities without causing significant divergence from the base models. Our proposed self-alignment with RRPO exhibits consistent improvements across all setups over the base models, effectively reducing hallucination and improving spatio-temporal reasoning, thus enabling safer and more reliable use of LVLMs. Moreover, we show that the approach scales well with more data and high-resolution temporal inputs, and generalizes well across diverse LVLM architectures and training setups. Future work can further investigate iterative self-alignment methodologies with RRPO, moving beyond the static dataset used in this work.

## Acknowledgment

We thank Ahmad Beirami for his valuable feedback and suggestions, which helped improve our paper. We also thank the Bank of Montreal and Mitacs for funding this research, and the Vector Institute for providing computational resources.

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

# Appendix

## A RRPO Pseudocode (PyTorch Style)

```python
import torch
import torch.nn.functional as F

def rrpo_loss(self, logits, ref_logits, phrase_ids, alpha, beta):
    """
    logits:         logits from pi_theta.
                    shape: (batch_size, sequence_length, vocab_size)
    ref_logits:     logits from pi_ref.
                    shape: (batch_size, sequence_length, vocab_size)
    phrase_ids:     phrase identifiers where tokens belonging to the same
                    phrase share the same value; additionally, they remain
                    the same between correct and misaligned phrases to
                    maintain correspondence.
                    shape: (batch_size, sequence_length)
    correct_idx:    indices of the correct responses in batch
    wrong_idx:      indices of the wrong responses in batch
    alpha:          coefficient to control the token-wise KL divergence
    beta:           coefficient to control reward/penalty
    """

    # compute log probabilities
    logps = logits.log_softmax(dim=-1)
    ref_ps = ref_logits.softmax(dim=-1)
    ref_logps = ref_ps.log()

    # compute token-wise KL divergence
    token_wise_kl = (ref_ps * (ref_logps - logps)).sum(dim=-1)

    # compute the margin
    logps_margin = logps - ref_logps

    # accumulate log probabilities of the phrases with key concepts
    # here 0 indicates ignored tokens
    unique_phrase_ids = torch.unique(phrase_ids, sorted=True)[1:]
    phrase_logps_margin = torch.zeros(
            phrase_ids.size(0), len(unique_phrase_ids))
    for i, phrase_id in enumerate(unique_phrase_ids):
        mask = (phrase_ids == phrase_id).float()
        phrase_logps_margin[:, i] = (logps_margin * mask).sum(dim=-1)

    chosen_logps_margin = phrase_logps_margin[correct_idx]
    rejected_logps_margin = phrase_logps_margin[wrong_idx]
    logits_margin = (chosen_logps_margin -
                        rejected_logps_margin).sum(dim=-1)

    chosen_token_wise_kl = token_wise_kl.sum(dim=-1)[chosen_idx]

    losses = -torch.logsigmoid(beta * logits_margin)
            + alpha * chosen_token_wise_kl

    return losses
```

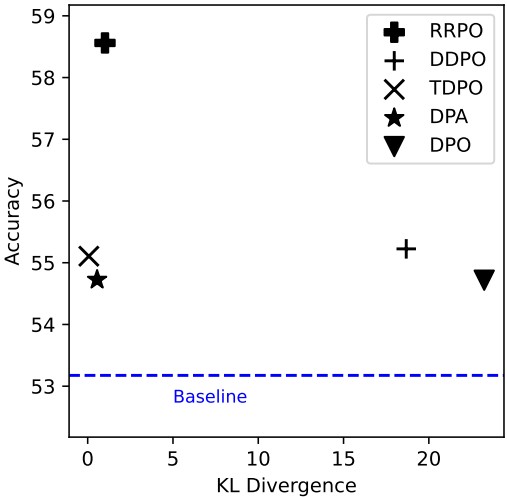

Figure S1: Comparison of characteristics among preference optimization methods. Both DDPO and DPO diverge significantly from their initial state after alignment. While TDPO and DPA are effective in restricting model divergence, they are less effective in improving performance. RRPO achieves excellent performance with minimal divergence from the base model.

## B    Comparing Loss Formulation of RRPO and Other Methods

This section presents a comparative analysis between RRPO and other preference optimization methods studied in this work, i.e., DPO, DDPO, TDPO, and DPA. We begin by outlining the respective loss functions, followed by a detailed discussion of their similarities and differences.

**Comparison with DPO.**

As previously discussed, the DPO [31] loss function is defined as:

$$
\begin{aligned}
\mathcal{L}_{\text{DPO}}(\pi_\theta; \pi_{\text{ref}}) &= -\mathbb{E}\bigg[\log \sigma\big(r_\theta(x, y^+) - r_\theta(x, y^-)\big)\bigg], \\
&= -\mathbb{E}\bigg[\log \sigma\Big(\beta \log \frac{\pi_\theta(y^+|x)}{\pi_{\text{ref}}(y^+|x)} - \beta \log \frac{\pi_\theta(y^-|x)}{\pi_{\text{ref}}(y^-|x)}\Big)\bigg].
\end{aligned}
\tag{S1}
$$

The DPO loss function calculates reward over all tokens in $y^+$ and $y^-$, despite the fact that there might be a few sub-sequences that are conceptually different. This approach results in coarse-grained reward modeling, and because it penalizes all tokens in the response, the loss function accumulates a large gradient and tends to diverge significantly from the base model, potentially resulting in weak alignment, as shown in Figure S1. RRPO is introduced to address two key challenges of DPO: to provide fine-grained feedback and restrict the divergence of the model away from its initial state.

**Comparison with DDPO.**

DDPO [53] extends DPO by incorporating a weighted reward based on the sub-sequence level differences between $y^+$ and $y^-$, and is defined as:

$$
\mathcal{L}_{\text{DDPO}}(\pi_\theta; \pi_{\text{ref}}) = -\mathbb{E}\bigg[\log \sigma\Big(\beta \log \frac{\pi_\theta(y^+|x)}{\pi_{\text{ref}}(y^+|x)} - \beta \log \frac{\pi_\theta(y^-|x)}{\pi_{\text{ref}}(y^-|x)}\Big)\bigg],
\tag{S2}
$$

where $\log \pi(y|x) = \sum_{y_i \in y} \log p(y_i|x, y_{<i})$ is modified as:

$$
\log \pi(y|x) = \frac{1}{N}\Big[\sum_{y_i \in y_{\text{same}}} \log p(y_i|x, y_{<i}) + \gamma \sum_{y_i \in y_{\text{different}}} \log p(y_i|x, y_{<i})\Big].
$$

Here, $y_{\text{same}}$ and $y_{\text{different}}$ indicate unchanged and changed segments between $y^+$ and $y^-$. Moreover, $\gamma > 1$ is a weighting hyperparameter, and larger $\gamma$ indicates more weight on those changed segments.

While DDPO is designed to provide fine-grained feedback, its loss formulation is not as effective as RRPO and is also prone to diverging far from its initial state, similar to DPO, due to weak regularization, see Figure S1.

**Comparison with TDPO.**

TDPO [34] is also derived from DPO, incorporating an additional regularization term ($\mathbb{D}_{\text{TKL}}$) between $\pi_\theta$ and $\pi_{\text{ref}}$, which is defined as:

$$
\begin{aligned}
\mathcal{L}_{\text{TDPO}}(\pi_\theta; \pi_{\text{ref}}) = - \mathbb{E}\Bigg[ &\log \sigma\bigg( \big(r_\theta(x, y^+) - r_\theta(x, y^-)\big) \\
&- \alpha\Big(\beta \mathbb{D}_{\text{TKL}}\left(x, y_w; \pi_{\text{ref}} \| \pi_\theta\right) - sg\big(\beta \mathbb{D}_{\text{TKL}}\left(x, y_c; \pi_{\text{ref}} \| \pi_\theta\right)\big)\Big)\bigg) \Bigg].
\end{aligned}
\tag{S3}
$$

As shown in Figure S1, TDPO is effective in restricting the divergence of the base model, but its performance is almost the same as DPO and falls short of our RRPO.

**Comparison with DPA.**

DPA [27] is a phrase-level alignment method unlike DPO and its variants. The DPA loss is composed of two terms where the first term computes the relative log-probability between two phrases of $y^+$ and $y^-$, and the second term works as a regularizer between $\pi_\theta$ and $\pi_{\text{ref}}$, formulated as:

$$
\mathcal{L}_{\text{DPA}}(\pi_\theta; \pi_{\text{ref}}) = - \mathbb{E}\left[ \frac{1}{N} \sum_{i=1}^{N} -\log \frac{P(y_i^+)}{P(y_i^+) + P(y_i^-)} + \alpha \cdot \mathbb{D}_{\text{TKL}}\big(x, y^+; \pi_{\text{ref}} \| \pi_\theta\big) \right],
\tag{S4}
$$

where $P(y_i^+)$ and $P(y_i^-)$ denote the probability of $i$-th phrase in $y^+$ and $y^-$. Assume, $y$ is expressed as a sequence of tokens $\{t_1, t_2, \ldots, t_{|T|}\}$, then the probability of $i$-th phrase can be computed as $\prod_{j=s_i}^{e_i} \pi_\theta(t_j | x, t_{<j})$, where $s_i$ and $e_i$ represent the start and end token indices.

The loss formulation of RRPO draws inspiration from DPA to achieve fine-grained alignment without the risk of divergence. However, we identify a fundamental limitation in DPA: it directly adjusts the probabilities of $\pi_\theta$ to modify the probability ratio between preferred and non-preferred phrases. This approach leads to an inaccurate probability ratio after the initial pair of preferred and non-preferred segments, as subsequent segment probabilities become dependent on their preceding elements. Therefore, DPA is not accurate in providing fine-grained feedback for sequences composed of multiple sub-sequences of key concepts. As shown in Figure S1, DPA, while successful in controlling divergence, is ineffective in improving performance.

# C  Additional Results

**Statistical analysis between RRPO and DPO.** We perform statistical analysis between RRPO and DPO aligned model. We consider the performance (denoted as `Score`) difference, whether an improvement or decline, of the `Model1` relative to the `Model2` on a specific task to be statistically significant if the Adjusted $\Delta$ exceeds zero, where $\Delta$ denotes the difference in performance between the `Model1` and `Model2`. Statistical significance is assessed using the Standard Error (SE) at a $95\%$ confidence level. The corresponding mathematical formulations are presented below.

$$\text{SE} = \sqrt{\frac{\texttt{Score}_{\texttt{Model1}} \times (1 - \texttt{Score}_{\texttt{Model2}})}{\text{number of samples}}},$$

$$\Delta = \texttt{Score}_{\texttt{Model1}} - \texttt{Score}_{\texttt{Model2}},$$

$$\text{Adjusted } \Delta = \Delta - 1.96 * \text{SE}.$$

The results presented in Table S1 shows that RRPO almost always outperforms DPO and even in several cases improvements are statistically significant.

Table S1: Statistical analysis between RRPO and DPO variants. Green underline indicates that the performance improvements are statistically significant. Also, there are no instances in which the DPO variants achieve statistically significant gains over RRPO.

| Model | TV Bench | TempCo-mpass$_{Avg}$ | Video Hallucer | Vid Halluc | MV Bench | Video MME | MLVU$_{val}$ (M-Avg) | LongVideo Bench$_{Val}$ |
|---|---|---|---|---|---|---|---|---|
| VideoChat2$_{7B}$+DPO | 45.7 | 60.0 | 22.1 | 72.4 | 59.7 | 43.0 | 47.4 | 41.0 |
| VideoChat2$_{7B}$+RRPO | 45.8 | 60.2 | 32.9 | 76.4 | 59.1 | 44.3 | 47.9 | 42.8 |
| LLaVA-Video$_{7B}$+DPO | 51.9 | 66.4 | 53.3 | 76.5 | 60.2 | 63.1 | 67.4 | 59.4 |
| LLaVA-Video$_{7B}$+RRPO | 51.9 | 66.8 | 55.7 | 76.5 | 62.0 | 64.5 | 69.1 | 60.4 |
| LLaVA-Video$_{7B}$+TPO | 51.1 | 66.1 | 50.6 | 76.3 | 60.6 | 65.6 | 68.9 | 60.1 |
| LLaVA-Video$_{7B}$+RRPO | 52.2 | 67.4 | 55.8 | 76.6 | 62.0 | 65.5 | 69.4 | 61.3 |
| LongVU$_{7B}$+DPO | 54.3 | 64.3 | 40.9 | 68.5 | 65.8 | 56.6 | 63.6 | 49.4 |
| LongVU$_{7B}$+RRPO | 56.5 | 64.5 | 44.0 | 71.7 | 66.7 | 57.7 | 64.5 | 49.7 |

**Detailed results.** This section details the results for the subcategories of the evaluation benchmarks used in our study.

Table S2: Detailed results on TempCompass.

| Models | Caption Matching | Captioning | Multi-choice | Yes-No | Avg. |
|---|---|---|---|---|---|
| VideoChat2$_{7B}$ | 69.5 | 46.6 | 58.0 | 63.0 | 59.3 |
| + RRPO | 73.2 | 48.5 | 56.6 | 62.6 | 60.2 |
| LlavaVideo$_{7B}$ | 75.1 | 50.2 | 67.6 | 71.0 | 66.0 |
| + RRPO | 75.8 | 52.0 | 68.6 | 70.9 | 66.8 |
| + RRPO (32f) | 76.6 | 53.0 | 68.7 | 71.3 | 67.4 |
| LongVU$_{7B}$ | 74.7 | 49.8 | 63.9 | 67.3 | 63.9 |
| + RRPO | 75.2 | 50.6 | 64.7 | 67.4 | 64.5 |

Table S3: Detailed results on VideoHallucer.

| Model | Object relation | Temporal | Semantic | Factual | Non-factual | Avg. |
|---|---|---|---|---|---|---|
| VideoChat2$_{7B}$ | 47.5 | 8.0 | 38.5 | 1.0 | 20.5 | 23.1 |
| + RRPO | 53.5 | 24.0 | 55.0 | 5.0 | 27.0 | 32.9 |
| LlavaVideo$_{7B}$ | 66.0 | 56.5 | 65.5 | 13.5 | 48.5 | 50.0 |
| + RRPO | 65.5 | 65.5 | 71.0 | 23.5 | 53.0 | 55.7 |
| + RRPO (32f) | 65.5 | 65.5 | 71.5 | 23.5 | 53.0 | 55.8 |
| LongVU$_{7B}$ | 50.5 | 46.0 | 43.0 | 17.0 | 39.5 | 39.2 |
| + RRPO | 53.0 | 48.0 | 50.0 | 26.0 | 43.0 | 44.0 |

Table S4: Detailed results on VidHalluc.

| Model | BinaryQA | MCQ | Scene Transition | Avg. |
|---|---|---|---|---|
| VideoChat2$_{7B}$ | 66.8 | 84.9 | 68.2 | 73.3 |
| + RRPO | 72.7 | 85.5 | 70.9 | 76.4 |
| LlavaVideo$_{7B}$ | 77.9 | 91.4 | 60.6 | 76.6 |
| + RRPO | 78.4 | 91.6 | 59.5 | 76.5 |
| + RRPO (32f) | 78.6 | 91.7 | 59.5 | 76.6 |
| LongVU$_{7B}$ | 71.4 | 87.0 | 43.4 | 67.3 |
| + RRPO | 74.2 | 88.2 | 52.7 | 71.7 |

Table S5: Detailed results on VideoMME.

| Model | Short | Medium | Long | Avg. |
|---|---|---|---|---|
| VideoChat2$_{7B}$ | 49.0 | 38.6 | 35.6 | 41.0 |
| + RRPO | 52.2 | 41.9 | 38.8 | 44.3 |
| LlavaVideo$_{7B}$ | 76.3 | 62.8 | 52.8 | 64.0 |
| + RRPO | 76.6 | 63.1 | 53.8 | 64.5 |
| + RRPO (32f) | 76.7 | 62.9 | 53.9 | 64.5 |
| LongVU$_{7B}$ | 66.1 | 54.7 | 47.9 | 56.2 |
| + RRPO | 67.7 | 55.0 | 50.3 | 57.7 |

# D   Additional Details of Training Data

**Prompt templates.**

The instructions used in processing open-ended generated responses employing GPT4o are presented in Figures S2 and S3.

```
% {Prompt used in open-ended response processing stage 1}
Thoroughly read the question and the given answers.

Your task is to determine whether the "Predicted answer" is "Correct" or "Wrong"
based on the "Question" and "Reference answer".
To determine correctness, focus on the key aspects in the answers, such as
objects, actions, and their attributes, among others.
The "Predicted answer" may have partial information in comparison to the
"Reference answer", in that case check whether at least the partial information
can be fully verified based on the "Reference answer".

Please respond with any of the following and nothing else:

- "Correct" if the predicted answer is correct based on the reference answer.
- "Wrong" if the predicted answer is not fully correct based on the reference
answer.
- "Undecided" if you are not sure about their correctness.

Question: {question}
Reference answer: {ground_truth}
Predicted answer: {generated_response}
```

Figure S2: Prompt used in open-ended response processing stage 1.

```
% {Prompt used in open-ended response processing stage 2}
***Turn 1***

Identify the key differences between these two sentences.
To identify differences focus on the key aspects in the sentences,
such as objects, actions, and their attributes, among others.
If there are no key difference between these two sentences,
please respond with "None" and nothing else.

Sentence 1: {sentence_from_ground_truth}
Sentence 2: {sentence_from_generated_response}

***Turn 2***

Please rewrite the "Sentence 1" by incorporating the differences you
mentioned earlier. Your final response should contain only the revised
sentence and nothing else.

Sentence 1: {sentence_from_ground_truth}
```

Figure S3: Prompt used in open-ended response processing stage 2.

# E Implementation Details

**Training hyperparameters.**

Table S6: Details of training hyperparameters.

|  | VideoChat2 | LLaVA-Video | LongVU |
|---|---|---|---|
| LLM | Mistral | Qwen2 | Qwen2 |
| Vision encoder | UMT | SigLIP | SigLIP+DINOv2 |
| Trainable module | LoRA in LLM and everything else is kept frozen | | |
| LoRA setup [50] | rank=128, alpha=256 | | |
| Learning rate | 2e-5 | 5e-6 | 5e-6 |
| Learning rate scheduler | Cosine | Cosine | Cosine |
| Optimizer | AdamW | AdamW | AdamW |
| Weight decay | 0.02 | 0.0 | 0.0 |
| Warmup ratio | - | 0.03 | 0.03 |
| Epoch | 1 | 1 | 1 |
| Batch size per GPU | 2 | 1 | 1 |
| Batch size (total) | 32 | 32 | 32 |
| $\alpha$ (loss coefficient) | 0.01 | 0.01 | 0.05 |
| $\beta$ (loss coefficient) | 0.9 | 0.1 | 0.5 |
| Memory optimization | - | Zero stage 3 [84, 85] | FSDP |

**Licenses of existing assets used.**

- VideoChat2 (Apache License 2.0): *https://huggingface.co/OpenGVLab/VideoChat2_stage3_Mistral_7B*

- LLaVA-Video (Apache License 2.0 ): *https://huggingface.co/lmms-lab/LLaVA-Video-7B-Qwen2*

- LongVU (Apache License 2.0): *https://huggingface.co/Vision-CAIR/LongVU_Qwen2_7B*

- VideoChat-IT (MIT): *https://huggingface.co/datasets/OpenGVLab/VideoChat2-IT*

# F Broader impact

As generative models are increasingly deployed in real-world applications, there is a growing need for post-training methods that enable fine-grained alignment with human preferences and values. Our proposed method, RRPO, can be applied to align generative models in both language and multimodal settings. By facilitating more precise alignment, RRPO has the potential to improve the safety, reliability, and usability of these models for real-world usage.

# G Qualitative Results

We present several examples in Figures S4 - S7 highlighting the effectiveness of RRPO over the base model and other preference optimization methods (e.g., DPO) in diverse video understanding tasks. We also present some failure cases in Figures S8 - S9.

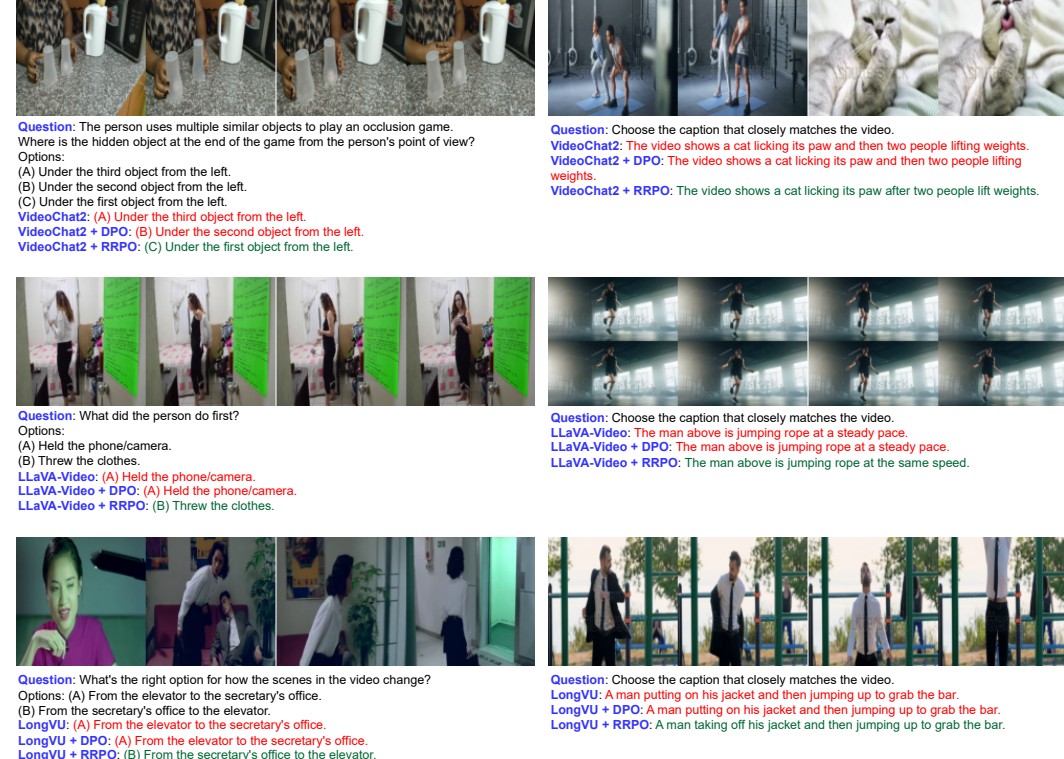

**Question**: The person uses multiple similar objects to play an occlusion game. Where is the hidden object at the end of the game from the person's point of view?
Options:
(A) Under the third object from the left.
(B) Under the second object from the left.
(C) Under the first object from the left.
**VideoChat2**: (A) Under the third object from the left.
**VideoChat2 + DPO**: (B) Under the second object from the left.
**VideoChat2 + RRPO**: (C) Under the first object from the left.

**Question**: Choose the caption that closely matches the video.
**VideoChat2**: The video shows a cat licking its paw and then two people lifting weights.
**VideoChat2 + DPO**: The video shows a cat licking its paw and then two people lifting weights.
**VideoChat2 + RRPO**: The video shows a cat licking its paw after two people lift weights.

**Question**: What did the person do first?
Options:
(A) Held the phone/camera.
(B) Threw the clothes.
**LLaVA-Video**: (A) Held the phone/camera.
**LLaVA-Video + DPO**: (A) Held the phone/camera.
**LLaVA-Video + RRPO**: (B) Threw the clothes.

**Question**: Choose the caption that closely matches the video.
**LLaVA-Video**: The man above is jumping rope at a steady pace.
**LLaVA-Video + DPO**: The man above is jumping rope at a steady pace.
**LLaVA-Video + RRPO**: The man above is jumping rope at the same speed.

**Question**: What's the right option for how the scenes in the video change?
Options: (A) From the elevator to the secretary's office.
(B) From the secretary's office to the elevator.
**LongVU**: (A) From the elevator to the secretary's office.
**LongVU + DPO**: (A) From the elevator to the secretary's office.
**LongVU + RRPO**: (B) From the secretary's office to the elevator.

**Question**: Choose the caption that closely matches the video.
**LongVU**: A man putting on his jacket and then jumping up to grab the bar.
**LongVU + DPO**: A man putting on his jacket and then jumping up to grab the bar.
**LongVU + RRPO**: A man taking off his jacket and then jumping up to grab the bar.

Figure S4: Qualitative examples on fine-grained temporal understanding tasks.

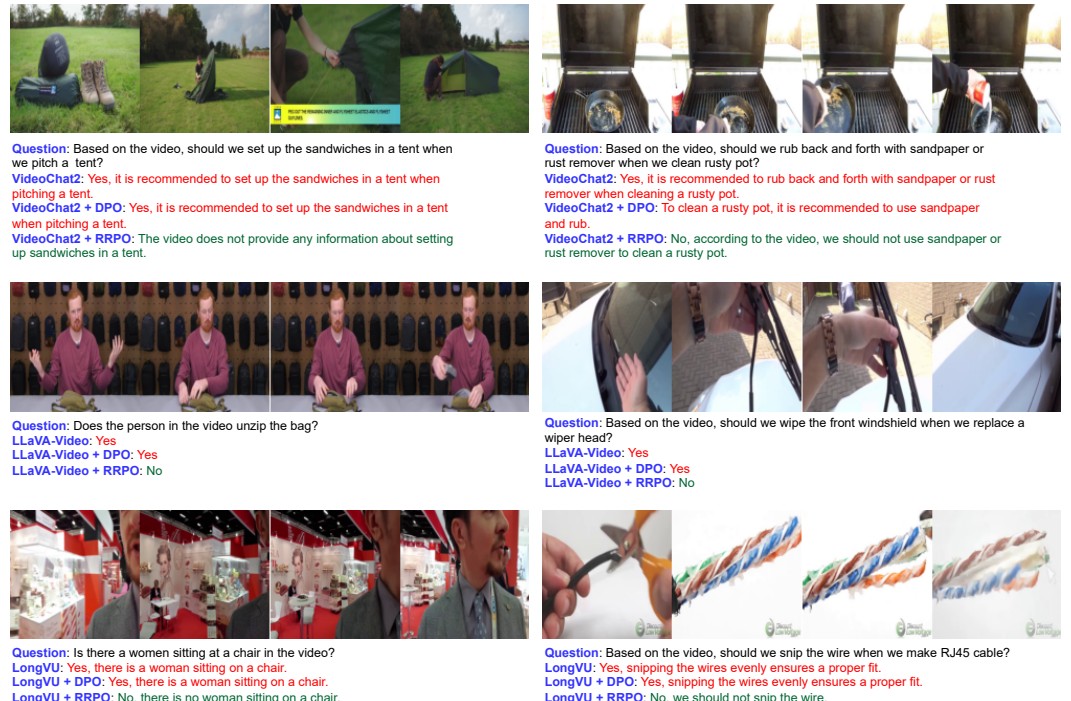

**Question**: Based on the video, should we set up the sandwiches in a tent when we pitch a tent?
**VideoChat2**: Yes, it is recommended to set up the sandwiches in a tent when pitching a tent.
**VideoChat2 + DPO**: Yes, it is recommended to set up the sandwiches in a tent when pitching a tent.
**VideoChat2 + RRPO**: The video does not provide any information about setting up sandwiches in a tent.

**Question**: Based on the video, should we rub back and forth with sandpaper or rust remover when we clean rusty pot?
**VideoChat2**: Yes, it is recommended to rub back and forth with sandpaper or rust remover when cleaning a rusty pot.
**VideoChat2 + DPO**: To clean a rusty pot, it is recommended to use sandpaper and rub.
**VideoChat2 + RRPO**: No, according to the video, we should not use sandpaper or rust remover to clean a rusty pot.

**Question**: Does the person in the video unzip the bag?
**LLaVA-Video**: Yes
**LLaVA-Video + DPO**: Yes
**LLaVA-Video + RRPO**: No

**Question**: Based on the video, should we wipe the front windshield when we replace a wiper head?
**LLaVA-Video**: Yes
**LLaVA-Video + DPO**: Yes
**LLaVA-Video + RRPO**: No

**Question**: Is there a women sitting at a chair in the video?
**LongVU**: Yes, there is a woman sitting on a chair.
**LongVU + DPO**: Yes, there is a woman sitting on a chair.
**LongVU + RRPO**: No, there is no woman sitting on a chair.

**Question**: Based on the video, should we snip the wire when we make RJ45 cable?
**LongVU**: Yes, snipping the wires evenly ensures a proper fit.
**LongVU + DPO**: Yes, snipping the wires evenly ensures a proper fit.
**LongVU + RRPO**: No, we should not snip the wire.

Figure S5: Qualitative examples on video hallucination tasks.

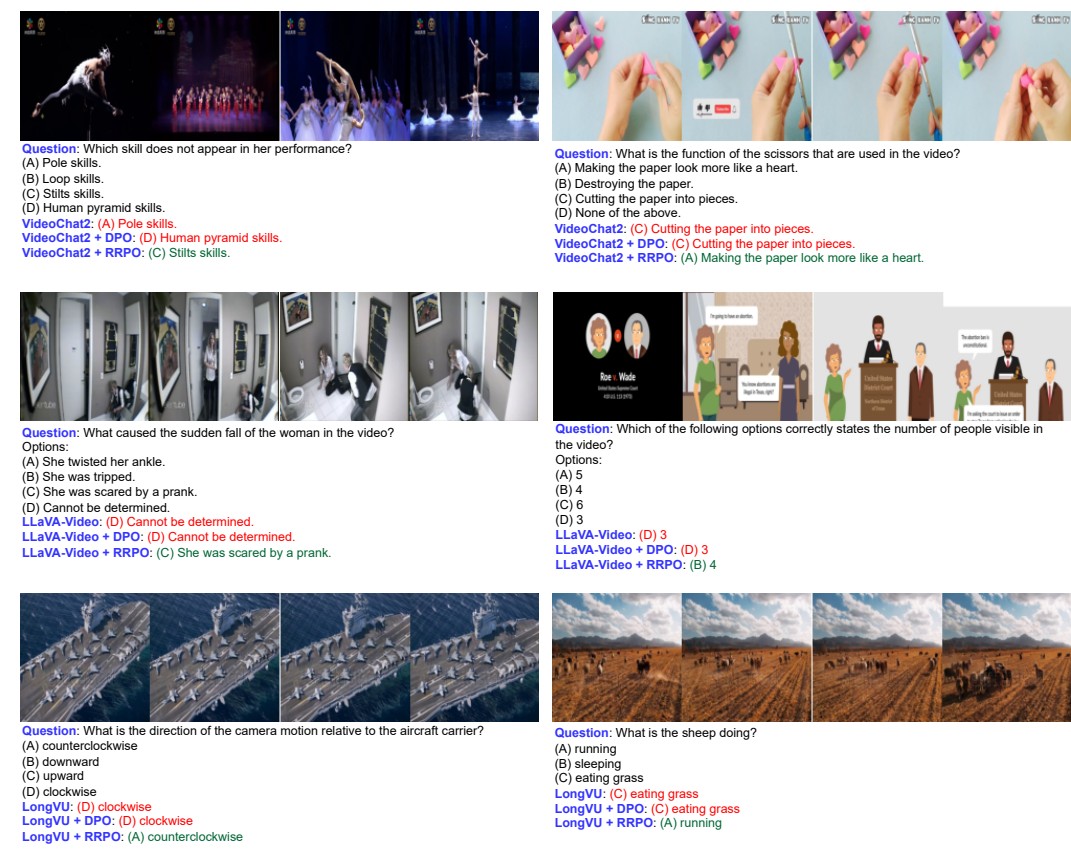

**Question**: Which skill does not appear in her performance?
(A) Pole skills.
(B) Loop skills.
(C) Stilts skills.
(D) Human pyramid skills.
**VideoChat2**: (A) Pole skills.
**VideoChat2 + DPO**: (D) Human pyramid skills.
**VideoChat2 + RRPO**: (C) Stilts skills.

**Question**: What is the function of the scissors that are used in the video?
(A) Making the paper look more like a heart.
(B) Destroying the paper.
(C) Cutting the paper into pieces.
(D) None of the above.
**VideoChat2**: (C) Cutting the paper into pieces.
**VideoChat2 + DPO**: (C) Cutting the paper into pieces.
**VideoChat2 + RRPO**: (A) Making the paper look more like a heart.

**Question**: What caused the sudden fall of the woman in the video?
Options:
(A) She twisted her ankle.
(B) She was tripped.
(C) She was scared by a prank.
(D) Cannot be determined.
**LLaVA-Video**: (D) Cannot be determined.
**LLaVA-Video + DPO**: (D) Cannot be determined.
**LLaVA-Video + RRPO**: (C) She was scared by a prank.

**Question**: Which of the following options correctly states the number of people visible in the video?
Options:
(A) 5
(B) 4
(C) 6
(D) 3
**LLaVA-Video**: (D) 3
**LLaVA-Video + DPO**: (D) 3
**LLaVA-Video + RRPO**: (B) 4

**Question**: What is the direction of the camera motion relative to the aircraft carrier?
(A) counterclockwise
(B) downward
(C) upward
(D) clockwise
**LongVU**: (D) clockwise
**LongVU + DPO**: (D) clockwise
**LongVU + RRPO**: (A) counterclockwise

**Question**: What is the sheep doing?
(A) running
(B) sleeping
(C) eating grass
**LongVU**: (C) eating grass
**LongVU + DPO**: (C) eating grass
**LongVU + RRPO**: (A) running

Figure S6: Qualitative examples on comprehensive video understanding tasks (short).

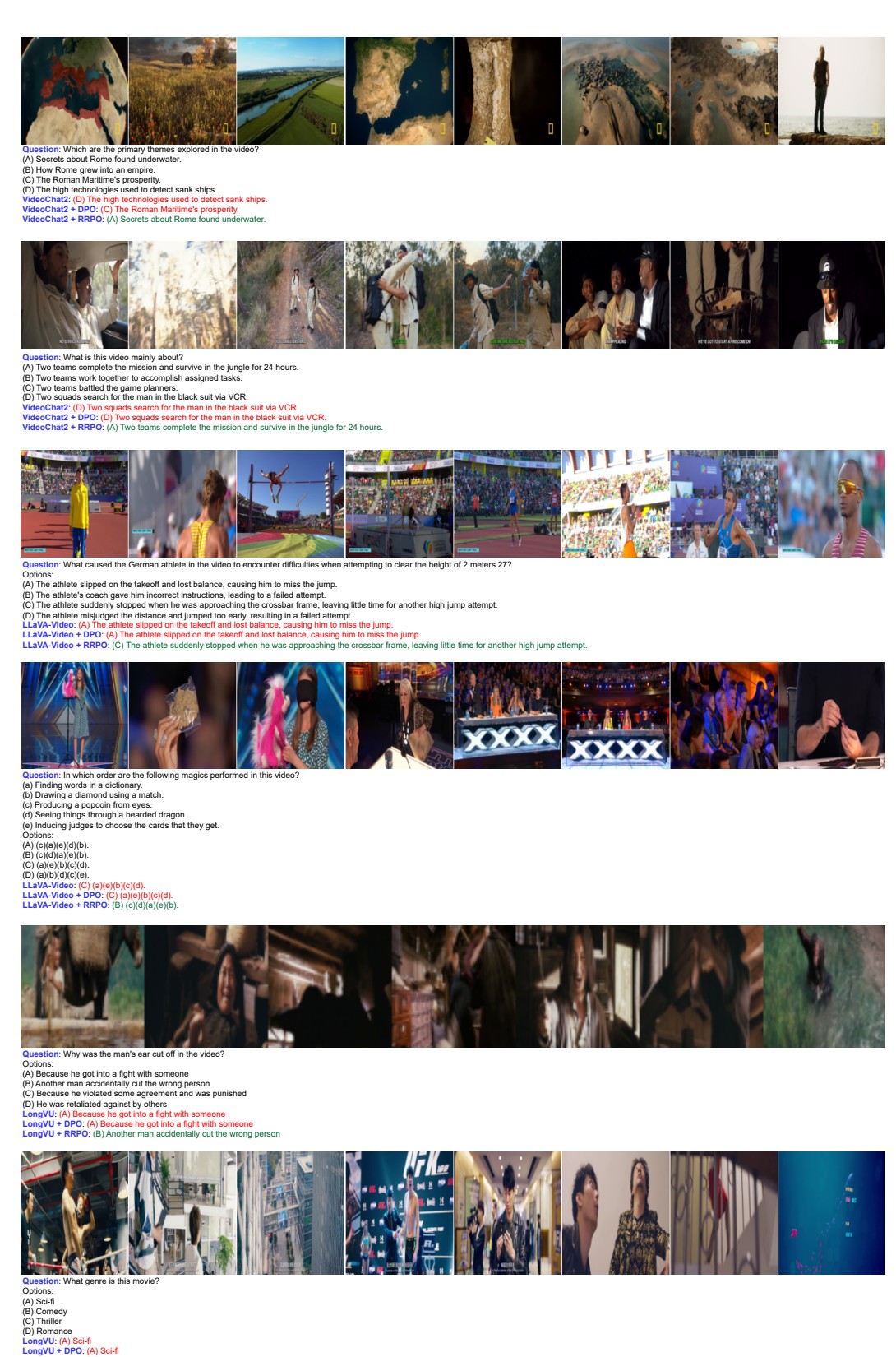

**Question**: Which are the primary themes explored in the video?
(A) Secrets about Rome found underwater.
(B) How Rome grew into an empire.
(C) The Roman Maritime's prosperity.
(D) The high technologies used to detect sank ships.
**VideoChat2**: (D) The high technologies used to detect sank ships.
**VideoChat2 + DPO**: (C) The Roman Maritime's prosperity.
**VideoChat2 + RRPO**: (A) Secrets about Rome found underwater.

**Question**: What is this video mainly about?
(A) Two teams complete the mission and survive in the jungle for 24 hours.
(B) Two teams work together to accomplish assigned tasks.
(C) Two teams battled the game planners.
(D) Two squads search for the man in the black suit via VCR.
**VideoChat2**: (D) Two squads search for the man in the black suit via VCR.
**VideoChat2 + DPO**: (D) Two squads search for the man in the black suit via VCR.
**VideoChat2 + RRPO**: (A) Two teams complete the mission and survive in the jungle for 24 hours.

**Question**: What caused the German athlete in the video to encounter difficulties when attempting to clear the height of 2 meters 27?
Options:
(A) The athlete slipped on the takeoff and lost balance, causing him to miss the jump.
(B) The athlete's coach gave him incorrect instructions, leading to a failed attempt.
(C) The athlete suddenly stopped when he was approaching the crossbar frame, leaving little time for another high jump attempt.
(D) The athlete misjudged the distance and jumped too early, resulting in a failed attempt.
**LLaVA-Video**: (A) The athlete slipped on the takeoff and lost balance, causing him to miss the jump.
**LLaVA-Video + DPO**: (A) The athlete slipped on the takeoff and lost balance, causing him to miss the jump.
**LLaVA-Video + RRPO**: (C) The athlete suddenly stopped when he was approaching the crossbar frame, leaving little time for another high jump attempt.

**Question**: In which order are the following magics performed in this video?
(a) Finding words in a dictionary.
(b) Drawing a diamond using a match.
(c) Producing a popcoin from eyes.
(d) Seeing things through a bearded dragon.
(e) Inducing judges to choose the cards that they get.
Options:
(A) (c)(a)(e)(d)(b).
(B) (c)(d)(a)(e)(b).
(C) (a)(e)(b)(c)(d).
(D) (a)(b)(d)(c)(e).
**LLaVA-Video**: (C) (a)(e)(b)(c)(d).
**LLaVA-Video + DPO**: (C) (a)(e)(b)(c)(d).
**LLaVA-Video + RRPO**: (B) (c)(d)(a)(e)(b).

**Question**: Why was the man's ear cut off in the video?
Options:
(A) Because he got into a fight with someone
(B) Another man accidentally cut the wrong person
(C) Because he violated some agreement and was punished
(D) He was retaliated against by others
**LongVU**: (A) Because he got into a fight with someone
**LongVU + DPO**: (A) Because he got into a fight with someone
**LongVU + RRPO**: (B) Another man accidentally cut the wrong person

**Question**: What genre is this movie?
Options:
(A) Sci-fi
(B) Comedy
(C) Thriller
(D) Romance
**LongVU**: (A) Sci-fi
**LongVU + DPO**: (A) Sci-fi
**LongVU + RRPO**: (B) Comedy

Figure S7: Qualitative examples on comprehensive video understanding tasks (long).

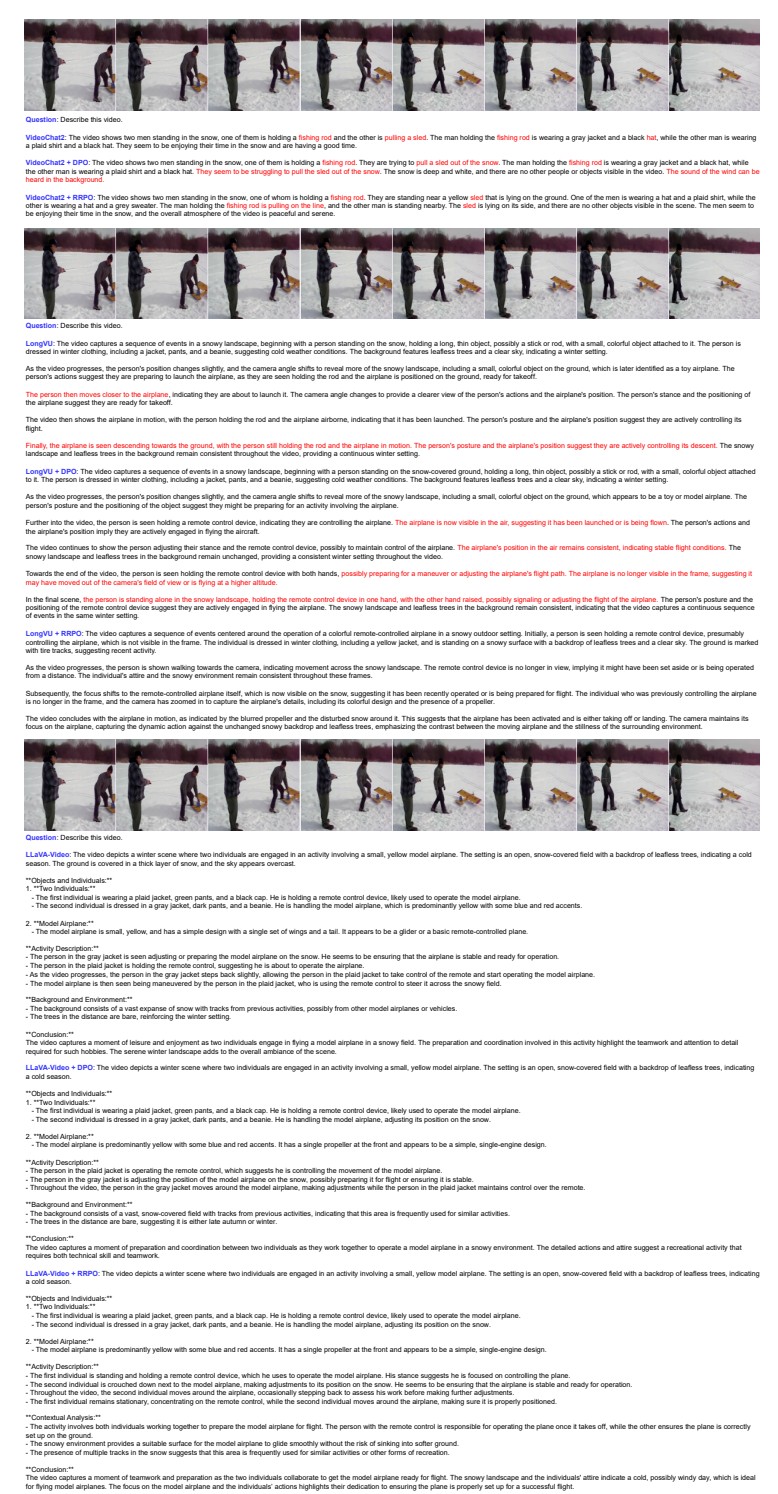

Figure S8: Building on the capabilities of the base models, we observe that RRPO-aligned models may still make mistakes or exhibit hallucinations in detailed video description tasks. For instance, VideoChat2 continues to display similar hallucinations after both DPO and RRPO training, as seen in the base model. In contrast, for LongVU, while both the base and DPO-aligned models hallucinate, the RRPO-aligned variant avoids such errors. Finally, in the case of LLaVA-Video, the RRPO-aligned model retains the base model's reliable behavior, as neither exhibits hallucinations.

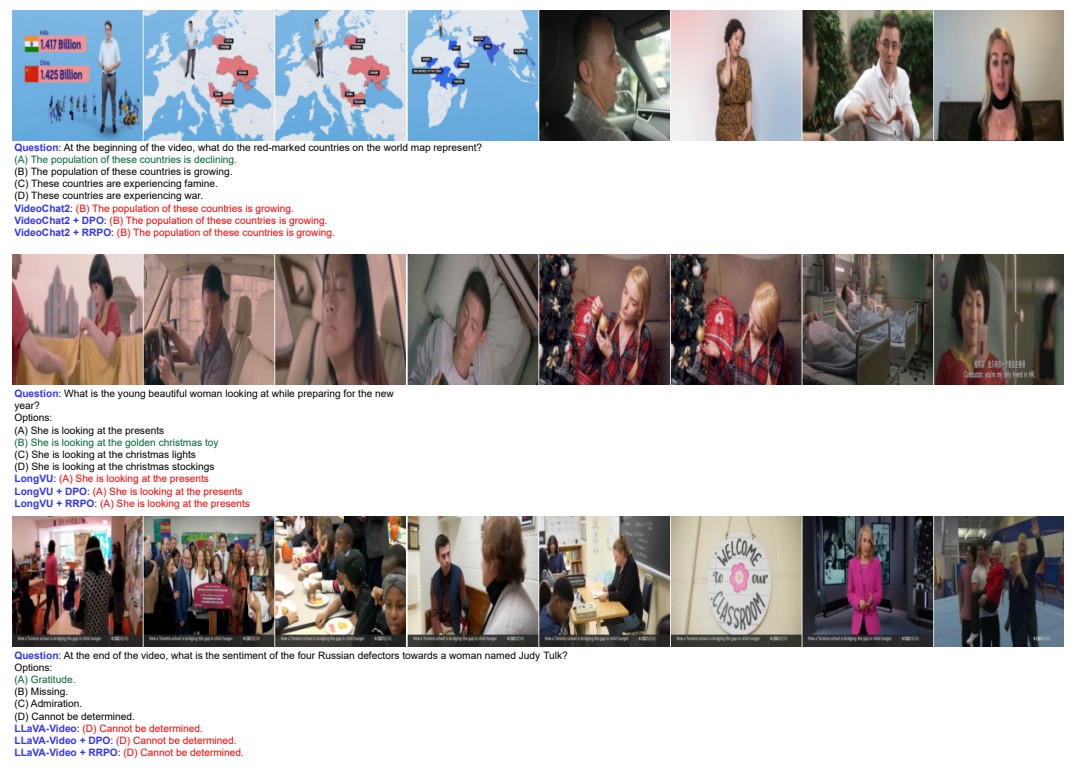

**Question**: At the beginning of the video, what do the red-marked countries on the world map represent?
(A) The population of these countries is declining.
(B) The population of these countries is growing.
(C) These countries are experiencing famine.
(D) These countries are experiencing war.
**VideoChat2**: (B) The population of these countries is growing.
**VideoChat2 + DPO**: (B) The population of these countries is growing.
**VideoChat2 + RRPO**: (B) The population of these countries is growing.

**Question**: What is the young beautiful woman looking at while preparing for the new year?
Options:
(A) She is looking at the presents
(B) She is looking at the golden christmas toy
(C) She is looking at the christmas lights
(D) She is looking at the christmas stockings
**LongVU**: (A) She is looking at the presents
**LongVU + DPO**: (A) She is looking at the presents
**LongVU + RRPO**: (A) She is looking at the presents

**Question**: At the end of the video, what is the sentiment of the four Russian defectors towards a woman named Judy Tulk?
Options:
(A) Gratitude.
(B) Missing.
(C) Admiration.
(D) Cannot be determined.
**LLaVA-Video**: (D) Cannot be determined.
**LLaVA-Video + DPO**: (D) Cannot be determined.
**LLaVA-Video + RRPO**: (D) Cannot be determined.

Figure S9: We also observe that RRPO-aligned models may still exhibit limitations in long video understanding tasks, primarily due to architectural constraints of the base model in processing extended frame sequences, as well as computational constraints during RRPO training that limit the use of long video inputs.

