# OpenReview forum: "Self-alignment of Large Video Language Models with Refined Regularized Preference Optimization"
_NeurIPS.cc/2025/Conference — NeurIPS 2025 poster_

### Official Review · Reviewer_baXK · 2025-06-28

**Clarity:** 4
**Significance:** 3
**Originality:** 4
**Rating:** 4
**Confidence:** 5

**Summary:**

This paper targets at improving Large Video Language Models (LVLMs) and propose a framework to have the model learn from the error they have made. In detail, the authors automatically generates pairs of preferred and non-preferred responses by introducing spatio-temporal perturbations into videos to simulate typical error patterns. Then, they designed Refined Regularized Preference Optimization, which is claimed to offer more precise alignment and more stable training, by utilizing reward at sub-sequence level. Both mathematical derivation and experimental validation has been provided to demonstrate the superiority of RRPO over DPO.

**Questions:**

Please see my weakness

**Ethical Concerns:**

["NO or VERY MINOR ethics concerns only"]

**Limitations:**

Please see the weakness

**Quality:**

4

**Strengths And Weaknesses:**

**Strength**:
- The motivation is important: learning from the errors to further enhance the model itself is important for application usage in the real world, in particular for fine-grained temporal understanding and reducing hallucination.
- The RRPO is novel and the idea of generating negative sequences by perturbing video is interesting.
- Detailed and extensive ablation study and discussion

**Weakness**:
- I am curious whether any capability of LVLM before RRPO is compromised. For example, what is the performance on the LVLM after RRPO on the cases that the model answered correctly before.
- Though I understand the claim regarding difference between coarse-grained and fine-grained level difference, but I am not sure whether this difference can be mitigated even with DPO. In other words, whether such difference really matters. Then, for the gradient scale (claim in Line 128), demonstration of the large gradient caused by long sentences is needed.
- Several important implementation details are missed, e.g., what is the value alpha used in the experiments.
- As part of future research direction, when the spatial & temporal perturbation is applied, the model prediction on some videos remains the same, which could be problematic. For example, when the key object for understanding is masked, if the model still makes the correct prediction, it can be treated as a hallucination, which is not preferred and should be mitigated. Then, how can your method be further improved to identifying such cases?

---

> ### Author Rebuttal · Authors · 2025-07-30
>
> We sincerely thank the reviewer for their time and thoughtful feedback. We are glad to see the overall positive remarks, particularly regarding the scope of this work for real-world applications and the novelty of our method. We have carefully addressed all points below and would be happy to clarify further if any questions remain.
>
> >**W1. What is the performance on the LVLM after RRPO on the cases that the model answered correctly before?**
>
> To evaluate this, we measure how many samples previously answered correctly by the base model remain correct after RRPO training. Let $B$ be the set of indices correctly answered by the base model, $R$ be the set correctly answered by the RRPO-trained model, and $T$ is the set of all indices in the evaluation set. We define retention\_score \= $100-(\frac{|B|}{|T|} - \frac{|B \\cap R|}{|T|}) \times 100$ .
>
> As shown below, both RRPO and DPO retain a very high percentage (97.2-97.9) of these samples across all benchmarks. However, the average improvement in performance in case of RRPO is much higher compared to DPO (2.26 vs. 0.46).
>
> | Method | Avg. retention\_score (%) | Avg. accuracy improved (%) |
> | :---- | :---- | :---- |
> | DPO | 97.9 | 0.46 |
> | **RRPO (Ours)** | 97.2 | 2.26 |
>
> >**W2. Does the difference between coarse-grained and fine-grained reward matter? Gradient comparison between DPO and RRPO.**
>
> **Difference between coarse-grained and fine-grained reward:** Fine-grained reward modeling is crucial for capturing subtle distinctions in response quality and correctness, especially when the model needs to understand why one response is preferred over another. This has far-reaching implications and is very crucial in domains where even minor errors can have significant consequences, for example, in medical, legal, and safety-critical applications.
>
> DPO provides only a coarse signal by treating the entire response as preferred or non-preferred. It does not indicate where the error occurred or which sub-sequence of the entire response contributed to the preference decision. As a result, the model receives limited guidance during alignment, which can lead to inefficiencies in learning. In contrast, our fine-grained reward modeling in RRPO explicitly identifies and localizes errors, providing more targeted supervision. As shown in Table 3 of the main paper, this leads to consistent and substantial improvements across all setups compared to DPO and other baselines, demonstrating the importance of fine-grained alignment.
>
> **Gradient comparison between DPO and RRPO; Claim in Line 128:** To support our claim in Line 128, we provide a theoretical analysis in Lines 149–174. Specifically, we show that the upper bound of the DPO gradient $\\nabla \_\\text{DPO}$ (see Eq. 10\) scales with the total length of the response pairs as $|y^+| \+ |y^-|$. In contrast, the upper bound of the RRPO gradient $\\nabla \_\\text{RRPO}$ (see Eq. 9\) scales with the length of the differing concepts i.e., $2NL$ where $N$  is the number of differing concepts between $y^+$ and $y^-$, and $L$  is the average token length of each differing concept.
> Since $|y^+| \+ |y^-| \>\> 2NL$, the upper bound of $\\nabla\_\\text{DPO}$ is significantly larger than that of   $\\nabla \_\\text{RRPO}$.
>
> Furthermore, the term $\\nabla\_{\\text{D}\_\\text{TKL}}$  in RRPO gradiebt is negative (see Line 167), which makes:
> $\\nabla\_{\\text{RRPO}} \<\< \\nabla\_{\\text{DPO}}$.
>
> We also provide empirical evidence in Figure 5, which shows that the large gradient in DPO leads to significant divergence from the model’s initial state, unlike RRPO. Importantly, **RRPO not only diverges less but also achieves better performance than DPO**, which is a highly desired trait at post-training.
>
> >**W3. Implementation details, e.g., value of $\\alpha$**
>
> All key implementation details, including training configurations and hyperparameters, are provided in Appendix E. As reported in Table S6, the value of $\\alpha$ was empirically set to 0.01 for VideoChat2 and LLaVA-Video, and 0.05 for LongVU. Please note that we also share the full codebase with default configurations to support reproducibility.
>
> >**W4. As part of future research direction, when the key object for understanding is masked, if the model still makes the correct prediction, it can be treated as a hallucination, which is not preferred and should be mitigated? Then, how can your method be further improved to identify such cases?**
>
> As a potential direction for future work, our method could be extended to identify and mitigate false positives arising from hallucinations. One possible approach is to adopt a guided masking strategy, where we explicitly track which objects or segments are masked. This would allow for more informed verification during training and help detect cases where the model relies on spurious cues. That said, implementing such a strategy may require richer annotations or additional visual grounding models during data generation, which introduces added complexity.

---

> > ### Author Response · Authors · 2025-08-03
> >
> > Dear Reviewer baXK,
> >
> > As we are towards the end of author-reviewer discussion phase, we respectfully inquire whether our responses have sufficiently addressed your comments.
> >
> > If so, we would be grateful if you would consider updating your score to reflect this. Should any concerns remain, we would be pleased to provide further clarification.
> >
> > Thank you again for your time and effort in reviewing our work.
> >
> > Best regards,\
> > Authors

---

> > ### Comment · Reviewer_baXK · 2025-08-05
> >
> > Thank you for your response. I have went through the reply and would like to keep my score as my questions are well-addressed

---

> > > ### Author Response · Authors · 2025-08-05
> > >
> > > We sincerely thank Reviewer baXK for acknowledging that our rebuttal addressed their concerns.

---

### Official Review · Reviewer_uoq4 · 2025-06-30

**Clarity:** 3
**Significance:** 2
**Originality:** 3
**Rating:** 5
**Confidence:** 4

**Summary:**

This work introduce Refined Regularized Preference Optimization (RRPO), a preference optimization method that utilizes sub-sequence-level refined rewards and token-wise KL regularization to address the limitations of DPO. The proposed method consistently improves performance over the base models and generalizes well across diverse training setups.

**Questions:**

See weakness.

If the authors can effectively resolve the issues I have highlighted, I would be willing to raise my rating accordingly.

**Ethical Concerns:**

["NO or VERY MINOR ethics concerns only"]

**Final Justification:**

The authors have addressed all my concerns and my final score is Accept.

**Limitations:**

yes

**Quality:**

3

**Strengths And Weaknesses:**

Strengths:
1. The proposed method shows promising improvements.
2. The experimental evaluation is well-executed.
3. The method consistently improves performance over the base models and generalizes well across diverse training setups.

Weaknesses:
1. The proposed method appears to be specifically designed for video understanding tasks. It would be helpful if the authors could clarify whether the approach is inherently limited to video inputs, or if it can be extended to image understanding tasks as well.

2. Lacks sufficient qualitative comparison results. Visual comparisons with baseline methods are essential to assess generation quality beyond quantitative metrics.

3. This work lacks an analysis of inference efficiency. It would be beneficial to include comparisons with baseline methods in terms of training time, GPU hours, and inference speed.

4. The method presents interesting results on low-resolution video datasets, and it would be valuable to explore the method’s performance in higher-resolution scenarios. Real-world applications often involve high-fidelity video inputs, which may introduce different challenges such as increased feature complexity or computational demands. It could be beneficial for the authors to discuss potential extensions to high-resolution data in the discussion section, or consider supplementary experiments on datasets with varied visual fidelity.

---

> ### Author Rebuttal · Authors · 2025-07-30
>
> We sincerely thank the reviewer for their time and thoughtful feedback. We are glad to see the overall positive remarks, particularly regarding the well-executed experimental setup and the generalizability of our approach. We have carefully addressed all points below and would be happy to clarify further if any questions remain.
>
>
> >**W1. Can our approach be extended to image tasks?**
>
> While our approach currently focuses only on videos, it can be easily adapted to image tasks with minor modifications. For example, in that case, only spatial perturbations (e.g., random masks to occlude parts of the image) would be applied to generate negative responses, due to the absence of a temporal dimension. In fact, as shown in our ablation study in Table 5 (main paper), spatial perturbation alone is also quite effective even on videos. The rest of the pipeline, including data preparation, reward formulation, and training procedure, remains largely unchanged.
>
> >**W2. Qualitative comparisons.**
>
> Qualitative examples are available in the root directory of our github repository (please see the link in the main paper) under a file named QualitativeComparisons.md. These examples further demonstrate the effectiveness of our method in comparison to the baselines on diverse video tasks including fine-grained temporal understanding, comprehensive video understanding, and hallucination. A few qualitative results are also provided in Fig. 1.
>
> >**W3. Training time comparisons and inference speed**
>
> We present a training time comparison of our method RRPO against other preference optimization baselines in the table below. We use LongVU-7B as the base model and train on a single node of 4 A100 80GB GPUs. As shown, the training times across methods are close to each other, demonstrating that training time required for RRPO is in the same range as others. Minor variations in training time can be attributed to fluctuations in data loading, server load, and other similar factors at the time of training.
>
> | Method | Training time (A100 80GB GPU hours) |
> | :---- | :---- |
> | DPO | 32 hrs. |
> | DPA | 33 hrs. |  |
> | TDPO | 32 hrs. |
> | DDPO | 32 hrs. |
> | **RRPO (Ours)** | 30 hrs. |
>
> We also evaluate the inference speed of all the methods. As expected, there is no difference across them, since none introduce architectural modifications that would impact runtime. We report inference speed under various generation lengths for completeness, in the table below.
>
> Response Length|Inference Speed|
> | :---- | :---- |
> |Short| 0.41 seconds per 10 tokens |
> |Medium| 2.95 seconds per 128 tokens |
> |Long| 6.70 seconds per 300 tokens |
>
> >**W4. Evaluation on higher-resolution scenarios**
>
> We would like to clarify that the evaluation benchmarks used in this work include videos with a wide range of spatial and temporal resolutions. Specifically, all benchmarks include videos with high pixel resolution (see table below). The temporal duration varies by task, for example, long-video understanding benchmarks feature videos of an hour long, while short-video tasks include videos of a few seconds to several minutes long. A summary of video characteristics is provided below. Table 3 in the main paper demonstrates strong performance across all these diverse benchmarks, indicating that our method is effective across various video resolutions (Res.) and durations (Dur.).
>
> | Benchmark | Res. Max (pixel) | Res. Min (pixel)  | Res. Avg (pixel)  | Dur. Max (sec.) | Dur. Min (sec.) | Dur. Avg (sec.) |
> | ----- | ----- | ----- | ----- | ----- | ----- | ----- |
> | TVBench | 1920 | 360 | 1150.36 | 116.02 | 1.24 | 18.84 |
> | TempCompass | 912 | 596 | 633.71 | 35.21 | 1.53 | 11.38 |
> | VideoHallucer | 1920 | 192 | 1073.78 | 623.02 | 0.2 | 80.63 |
> | VidHalluc | 3414 | 160 | 929.53 | 228.49 | 0.6 | 25.47 |
> | MVBench | 1920 | 224 | 907.39 | 116.02 | 1.03 | 15.93 |
> | VideoMME | 1280 | 480 | 1195.26 | 3579.45 | 11.03 | 1021.27 |
> | MLVU | 2922 | 320 | 1183.84 | 32550.1 | 179.97 | 943.02 |
> | LongVideo | 1280 | 674 | 1241.52 | 3573.96 | 7.97 | 459.51 |

---

> > ### Author Response · Authors · 2025-08-03
> >
> > Dear Reviewer uoq4,
> >
> > As we are towards the end of author-reviewer discussion phase, we respectfully inquire whether our responses have sufficiently addressed your comments.
> >
> > If so, we would be grateful if you would consider updating your score to reflect this. Should any concerns remain, we would be pleased to provide further clarification.
> >
> > Thank you again for your time and effort in reviewing our work.
> >
> > Best regards,\
> > Authors

---

> > ### Comment · Reviewer_uoq4 · 2025-08-05
> >
> > Thanks for your detailed response. All my concerns have been resolved.

---

> > > ### Author Response · Authors · 2025-08-05
> > >
> > > We sincerely thank Reviewer uoq4 for acknowledging that our rebuttal addressed their concerns.

---

### Official Review · Reviewer_jPTL · 2025-07-01

**Clarity:** 3
**Significance:** 3
**Originality:** 3
**Rating:** 4
**Confidence:** 4

**Summary:**

This paper proposes a self-alignment framework for Large Video Language Models (LVLMs), where models learn from their own errors via responses to perturbed videos. The key contribution is Refined Regularized Preference Optimization (RRPO), a novel training objective that combines sub-sequence-level rewards with token-wise KL regularization. RRPO improves fine-grained temporal reasoning, reduces hallucinations, and achieves consistent gains over prior alignment methods like DPO, as validated across multiple LVLM architectures and video understanding benchmarks.

**Questions:**

1. The paper uses a GPT-4o-mini model to assess correctness for open-ended tasks. Is there a risk of bias or alignment drift if this LLM acts as both the judgment oracle and learning target?
2. The method relies on identifying key concept pairs (y+i, y-i) for sub-sequence reward calculation. How robust is this identification process when scaling to noisy or less structured video-language datasets?

**Ethical Concerns:**

["NO or VERY MINOR ethics concerns only"]

**Final Justification:**

The paper introduces a novel self-alignment framework for LVLMs with RRPO, which effectively improves temporal reasoning, reduces hallucinations, and outperforms previous methods. All my concerns have been addressed.

**Limitations:**

Yes

**Quality:**

3

**Strengths And Weaknesses:**

Strengths:
- Technical Quality: The paper presents a rigorously designed training framework with clear theoretical and empirical justification.
- Empirical Rigor: The authors evaluate their method across a broad spectrum of benchmarks covering temporal reasoning, hallucination mitigation, and long-video understanding. RRPO consistently improves over baseline and existing methods with performance gains up to 5.4%.
- Innovation: The RRPO method combines fine-grained sub-sequence-level preference modeling and token-level KL regularization, addressing key weaknesses of DPO (e.g., coarse reward modeling, training instability).
- Scalability & Automation: The self-alignment data generation pipeline is automatic and does not require human annotation, allowing scalable construction of (preferred, non-preferred) response pairs from perturbed video samples.

Weaknesses:
- Clarity and Writing Quality: While the technical content is solid, the presentation can be improved. Some notations are densely packed, and explanations (e.g., around Eq. 6–8) require careful reading to interpret. Moreover, acronyms like RS-Mask, LS-Mask, etc., appear in Table 5 without early clear definitions (though defined in text).
- Formatting & Style: The manuscript contains minor formatting inconsistencies (e.g., spacing issues in equations and figure captions, long caption texts), which reduce readability.

---

> ### Author Rebuttal · Authors · 2025-07-30
>
> We sincerely thank the reviewer for their time and thoughtful feedback. We are pleased to see the positive remarks, particularly on the rigorous empirical and theoretical analysis, the innovative fine-grained reward modeling, and the scalability of our proposed approach. We have carefully addressed all points below, please let us know if any questions remain.
>
> >**W1. Densely packed mathematical notations and acronyms.**
>
> Due to space constraints, some mathematical expressions are written in-line, which we acknowledge can reduce readability. We will use the additional page allowed in the camera-ready version to reformat these expressions into separate lines wherever possible to improve clarity.
>
> Regarding the acronyms RS (Random Shuffling), LS (Local Shuffling), and GS (Global Shuffling), their definitions are provided in Lines 190–193 under Section 4\. Following the reviewer’s suggestion, we will revise the text to make such acronyms clearer and easier to read.
>
> >**W2. Formatting & Style**
>
> We thank the reviewer for their attention to detail. We will ensure that these minor formatting inconsistencies are corrected in the camera-ready version.
>
> >**Q1. Risk of bias or alignment drift due to the usage of GPT-4o?**
>
> We would like to clarify that GPT-4o is *not* used as the learning target, so concerns about bias or alignment drift do not apply in our setting. To illustrate the process, consider a video captioning example provided in Fig. 4 of the main paper. Suppose the ground-truth response of a video is: *“The video features a group of people playing soccer on a sandy field.”* Now assume the LVLM generates the response as: *“A group of people is seen playing volleyball on a beach covered in sand.”*. We use GPT-4o to extract the key *incorrect concepts* from the LVLM-generated response, in this case, “volleyball” vs. “soccer” and “beach” vs. “field”. We then rewrite the *ground-truth response* by incorporating these incorrect concepts to produce a closely *aligned* *non-preferred response* (which are used at fine-grained alignment training) such as: *“The video shows a group of people playing volleyball on a sandy beach.”*
>
> In this way, the model is aligned with the ground-truth or preferred responses, which are obtained from the instruction-tuning dataset while explicitly contrasting its own mistakes. Thus, GPT-4o serves only as a tool for identifying and manipulating semantic differences, not as a learning target. Further details on the instructions used in processing the responses are provided in Appendix D.
>
> >**Q2. Robustness of the identification of incorrect concepts when scaling to noisy or less structured video-language datasets?**
>
> We find that identifying key concept pairs $(y^+, y^-)$ using LLMs is fairly robust in practice, particularly as current LLMs such as GPT4o variants have become increasingly capable at handling such language understanding tasks. To further ensure reliability of this process, we extract incorrect concepts sentence by sentence, rather than treating them as monolithic texts. Our method successfully processes responses from a range of diverse video-language datasets, including VideoChat, VideoChatGPT, EGO-QA, TGIF-QA, and WebVid-QA, demonstrating its generalizability. While we acknowledge that LLMs can still make occasional errors in this identification process, this remains the most practical and scalable approach. Manual annotation or human-in-the-loop revision, though potentially more accurate, would be prohibitively expensive and infeasible at scale. Given these trade-offs, our approach offers a strong balance between scalability and perfection.

---

> > ### Author Response · Authors · 2025-08-03
> >
> > Dear Reviewer jPTL,
> >
> > As we are towards the end of author-reviewer discussion phase, we respectfully inquire whether our responses have sufficiently addressed your comments.
> >
> > If so, we would be grateful if you would consider updating your score to reflect this. Should any concerns remain, we would be pleased to provide further clarification.
> >
> > Thank you again for your time and effort in reviewing our work.
> >
> > Best regards,\
> > Authors

---

> > > ### Comment · Reviewer_jPTL · 2025-08-06
> > > **Concerns Addressed**
> > >
> > > Thank you for the detailed response. Most of my concerns have been addressed, and I appreciate the clarifications provided.

---

> > > > ### Author Response · Authors · 2025-08-06
> > > >
> > > > We sincerely thank Reviewer jPTL for acknowledging that our rebuttal sufficiently addressed their concerns.

---

### Official Review · Reviewer_xves · 2025-07-03

**Clarity:** 4
**Significance:** 2
**Originality:** 2
**Rating:** 5
**Confidence:** 3

**Summary:**

This work proposes a self-alignment framework for large vision-language models called RRPO, which enables them to learn from their own errors without any human-annotated preferences. Rather than relying on manually labeled preference pairs, RRPO generates spatial and temporal perturbations of video inputs, uses an external LLM to score and rewrite outputs, and applies sub-sequence-level token-wise KL-regularized rewards. RRPO demonstrates significant gains on hallucination reduction, temporal reasoning, and both short- and long-video understanding benchmarks across multiple LVLM architectures. It also provides tight gradient-norm bounds that justify its stable, efficient training and allow higher learning rates.

**Questions:**

My main concerns are in the method section, please see the weakness above.

Additionally:
1. Does the fine-grained focus and reward-hacking token-wise KL regularizer result in a trade-off between hallucination suppression and general language-vision capabilities, where the model overfits to hallucination-related metrics, achieving high scores without genuinely improving visual grounding, since the gains are primarily on hallucination benchmarks?

**Ethical Concerns:**

["NO or VERY MINOR ethics concerns only"]

**Final Justification:**

Strengths:
1. The paper is well-written and easy to understand.
2. The fine-grained motivation is interesting and appears to complement DPO’s coarse-grained training approach.
4. Comprehensive evaluation across three diverse LVLMs and six benchmarks (temporal reasoning, hallucination, short- and long-video understanding) demonstrates consistent performance gains.
5. Theoretical analysis about RRPO’s gradient norm justifying its stability during model training.
6. Code, data, and model weights are provided to ensure reproducibility.

After rebuttal:
I was very impressed and satisfied with how authors addressed my concerns regarding experiments on failure cases or domain shifts. Others questions are also clearly resolved, thus I decided to raise my score.

**Limitations:**

The discussion of this work's limitations seems brief and insufficient, as it does not properly address the failure modes of the approach or the trade-off between fine-grained and coarse-grained learning.

**Quality:**

3

**Strengths And Weaknesses:**

Strengths:
1. The paper is well-written and easy to understand.
2. The fine-grained motivation is interesting and appears to complement DPO’s coarse-grained training approach.
4. Comprehensive evaluation across three diverse LVLMs and six benchmarks (temporal reasoning, hallucination, short- and long-video understanding) demonstrates consistent performance gains.
5. Theoretical analysis about RRPO’s gradient norm justifying its stability during model training.
6. Code, data, and model weights are provided to ensure reproducibility.

Weaknesses:
1. Synthetic perturbations may not accurately capture real-world error distributions, risking overfitting to artificial failure modes. Since this strategy is not explicitly tied to actual model failures, it can lead to a distribution misalignment between the generated negatives and the model’s true hallucination behavior.
2. The requirement for non-preferred responses to maintain structural similarity with the original responses may hinder the model’s ability to generalize to diverse response patterns.
3. Experiments on failure cases or domain shifts are not thoroughly analyzed.
4. The discussion of this work's limitations seems brief and insufficient.

---

> ### Author Rebuttal · Authors · 2025-07-30
>
> We sincerely thank the reviewer for their time and thoughtful feedback. We are pleased to see the overall positive remarks, especially regarding the clarity of writing, the motivation behind fine-grained reward modeling, the comprehensive evaluations, and the theoretical analysis. We have carefully addressed all points below, please let us know if any questions remain.
>
> >**W1. Can synthetic perturbations in generating negative responses capture the model’s true hallucination behavior?**
>
> Our experiments on the data generation pipeline  (see Sec. 5, Q7) show that mild perturbations are particularly effective in generating negative responses that capture true hallucination and error patterns without introducing unrealistic artifacts such as overfitting to artificial failure modes or misaligned negative responses. As a result, our method improves accuracy by 4.8% to 9.8% on hallucination tasks. Our study further reveals that the absence of any perturbations leads to diminished model performance due to poor generalizability (please see Table 5).
>
> >**W2. Could the requirement for non-preferred responses to maintain structural similarity with the ground-truth responses (for fine-grained alignment) limit the model’s generalizability to diverse response patterns?**
>
> Maintaining structural similarity is not restricted to only the non-preferred responses. In principle, we could rephrase the ground-truth responses to match the semantic structure of the non-preferred ones for fine-grained alignment. However, we adopted a more conservative approach, keeping the ground-truths fixed and modifying only the non-preferred responses, to avoid potentially noisy alignment targets. To further enrich error diversity multiple responses can also be obtained for one original sample with varying perturbations and sampling temperatures. We avoided this direction in order to stay within a fixed computational budget, where we opted for sample diversity across instances rather than generating multiple variants per instance. Our results in Table 3 show that our approach for generating negative responses is effective and generalizes well across a variety of video tasks including comprehensive short- and long-video understanding, hallucination tasks, and fine-grained temporal reasoning.
>
> >**W3. Experiments on failure cases or domain shifts.**
>
> **Failure cases**: Examples of failed cases are available in the root directory of our github repository (please see the link in the main paper) under a file named FailedCases.md. We notice that depending on the capabilities of the base models, RRPO-aligned models can still make mistakes or exhibit hallucinations in detailed video description tasks. We also observe that RRPO-aligned models may still exhibit limitations in long video understanding tasks, primarily due to architectural constraints of the base model in processing extended frame sequences, as well as computational constraints during RRPO training that limit the use of long video inputs.
>
> **Domain shift**: Given that the underlying general-purpose LVLMs are trained on massive web-scale corpora during their pre-training and post-training phase, including millions or billions of diverse videos and images, and trillions of language tokens, drawing a clear boundary around what constitutes a “shift” is non-trivial. Moreover, in many cases, the details of the pre-training datasets are not public. Nevertheless, we believe that the diverse set of benchmarks we evaluate on, spanning a wide range of video tasks including video hallucination, comprehensive short- and long-video understanding, and fine-grained temporal reasoning, provides a strong proxy for robustness under diverse conditions.
>
> >**W4. Discussion on limitations.**
>
> Our RRPO method introduces fine-grained reward modeling, which requires preference pairs to maintain structural symmetry, unlike the coarse-grained approach used in DPO, which imposes no such constraint. While fine-grained alignment requires an additional step in data preparation, modern LLMs (e.g., variants of GPT-4o, Gemini, LLaMA, Qwen) are increasingly adept at handling such language tasks, making this step relatively easy to automate. More importantly, given the strong performance gains achieved through our fine-grained alignment strategy, we believe this additional step is a worthwhile trade-off. We will include this discussion in the final version of our paper.
>
> >**Q1. Is there a trade-off between hallucination suppression and general language-vision capabilities?**
>
> While the gains on hallucination benchmarks are more pronounced than those on non-hallucination tasks, we do not observe a trade-off where reducing hallucinations comes at the cost of general capabilities. In fact, the fine-grained reward modelling in RRPO loss is explicitly designed to suppress hallucinations without compromising overall language-vision performance, unlike DPO. As shown in the table below, RRPO improves performance in both hallucination (VideoHallucer) and general benchmarks (MVBench, VideoMME, MLVU) while DPO improves performance  (while lower than RRPO) on hallucination benchmarks but leads to a drop in general capabilities.
>
> | Model | VideoHallucer $\\uparrow$ | MVBench $\\uparrow$ | VideoMME $\\uparrow$ | MLVU $\\uparrow$ |
> | :---- | :---- | :---- | :---- | :---- |
> | LLaVA-Video 7B | 50.0 | 61.1  | 64.0 | 68.6 |
> | LLaVA-Video 7B \+ DPO | 53.3 | 60.6  | 63.1 | 67.4 |
> | LLaVA-Video 7B \+ **RRPO** (**Ours**) | 55.7 | 62.2  | 64.5 | 69.1 |

---

> > ### Author Response · Authors · 2025-08-03
> >
> > Dear Reviewer xves,
> >
> > As we are towards the end of author-reviewer discussion phase, we respectfully inquire whether our responses have sufficiently addressed your comments.
> >
> > If so, we would be grateful if you would consider updating your score to reflect this. Should any concerns remain, we would be pleased to provide further clarification.
> >
> > Thank you again for your time and effort in reviewing our work.
> >
> > Best regards,\
> > Authors

---

> > > ### Author Response · Authors · 2025-08-06
> > >
> > > Dear Reviewer xves,
> > >
> > > Given the limited time remaining in the author–reviewer discussion phase, we would be grateful if you could kindly confirm whether our rebuttal has sufficiently addressed your comments. If any concerns remain, we would be happy to provide further clarification.
> > >
> > > Best regards,\
> > > Authors

---

> > ### Comment · Reviewer_xves · 2025-08-06
> > **Rebuttal acknowledged**
> >
> > Thanks for authors' detailed and timely rebuttal. All of my questions have been answered, and I am happy to raise my score.

---

> > > ### Author Response · Authors · 2025-08-06
> > >
> > > We sincerely thank Reviewer xves for acknowledging that our rebuttal addressed all of their concerns and for accordingly increasing their scores.

---

### Note · Authors · 2025-08-11

Dear Review Committee,

We sincerely thank you for your time and effort in reviewing our paper and for providing insightful feedback. We are encouraged to note that **all reviewers** recognized the value of our work and **recommended acceptance** in their initial reviews.

The reviewers highlighted several strengths of our paper, including:

* Clarity of writing
* Strong motivation for fine-grained reward modeling
* Comprehensive and rigorous evaluations
* Theoretical analysis
* Scalability and Generalizability of the proposed approach
* Applicability to real-world scenarios
* Novelty of the method

During the rebuttal, we carefully addressed all comments shared by the reviewers. We are pleased to see that our responses are well received, with **all reviewers acknowledging that their concerns have been resolved and most indicating an increase in their initial scores**.

A brief summary of the key points discussed during the rebuttal is as follows:

* In response to Reviewers xves and jPTL, we highlight the robustness and effectiveness of our data generation pipeline in producing negative responses that capture true error patterns without introducing unrealistic artifacts in diverse tasks, essentially resulting in improved performance.
* In response to Reviewer xves, we further discuss the effectiveness of RRPO in mitigating harmful behavior (e.g., hallucination), while also improving general capabilities (e.g., fine-grained temporal understanding, comprehensive video understanding across varying video lengths) – a highly desired trait often missing in existing preference optimization methods.
* In response to Reviewers uoq4 and baXK, we discuss the possibility of expanding our proposed method beyond video domains, along with other potential future directions.
* In response to Reviewer uoq4, we highlight that the training time for RRPO is in the same range with existing preference optimization methods, and there is no increase in inference time as our method does not involve any architectural changes in the base model.
* Moreover, in response to Reviewer baXK, we discuss the benefits of fine-grained reward modeling in RRPO, such as: i) stable gradient updates, ii) faster convergence, iii) minimal divergence post-alignment, and iv) more effective alignment overall.

Thank you once again for reviewing our work and for providing such valuable feedback, which resulted in improving our paper.

---

### Decision · Program_Chairs · 2025-09-17

**Decision:**

Accept (poster)

**Comment:**

This paper improves the DPO algorithm by introducing sub-sequence–level rewards and token-wise KL regularization. Experiments across multiple LVLM architectures and video understanding benchmarks demonstrate strong empirical gains. The authors have satisfactorily addressed reviewer concerns, and these clarifications should be incorporated into the final version. As a general enhancement to DPO, it would be interesting to explore the method’s applicability beyond video understanding in future work.